# Aligning Embeddings and Geometric Random Graphs: Informational Results and Computational Approaches for the Procrustes-Wasserstein Problem

**Mathieu Even**
DI ENS, CRNS, PSL University, INRIA Paris

**Luca Ganassali**
Université Paris-Saclay, LMO

**Jakob Maier**
D.I ENS, CRNS, PSL University, INRIA Paris

**Laurent Massoulié**
D.I ENS, CRNS, PSL University, INRIA, MSR-INRIA Joint Centre, Paris

## Abstract

The Procrustes-Wasserstein problem consists in matching two high-dimensional point clouds in an unsupervised setting, and has many applications in natural language processing and computer vision. We consider a planted model with two datasets $X, Y$ that consist of $n$ datapoints in $\mathbb{R}^d$, where $Y$ is a noisy version of $X$, up to an orthogonal transformation and a relabeling of the data points. This setting is related to the graph alignment problem in geometric models. In this work, we focus on the euclidean transport cost between the point clouds as a measure of performance for the alignment. We first establish information-theoretic results, in the high ($d \gg \log n$) and low ($d \ll \log n$) dimensional regimes. We then study computational aspects and propose the 'Ping-Pong algorithm', alternatively estimating the orthogonal transformation and the relabeling, initialized via a Franke-Wolfe convex relaxation. We give sufficient conditions for the method to retrieve the planted signal after one single step. We provide experimental results to compare the proposed approach with the state-of-the-art method of Grave et al. [2019].

## 1 Introduction

Finding an alignment between high dimensional vectors or across two point clouds of embeddings has been the focus of recent threads of research and has a variety of applications in computer vision, such as inferring scene geometry and camera motion from a stream of images [Tomasi and Kanade, 1992], as well as in natural language processing such as automatic unsupervised translation [Rapp, 1995, Fung, 1995].

Many practical algorithms proposed for this task view this problem as minimizing the distance across distributions in $\mathbb{R}^d$. Some approaches are based e.g. on optimal transport and Gromov-Wasserstein distance Alvarez-Melis and Jaakkola [2018] or adversarial learning [Zhang et al., 2017, Conneau et al., 2018]. Another line of methods adapt the iterative closest points procedure (ICP) – originally introduced in Besl and McKay [1992] for 3-D shapes – to higher dimensions Hoshen and Wolf [2018]. Another recent contribution is that of Grave et al. [2019], where a method is proposed to jointly learn an orthogonal transformation and an alignment between two point clouds by alternating the objectives in the corresponding minimization problem.

38th Conference on Neural Information Processing Systems (NeurIPS 2024).

To formalize this problem, we consider a Gaussian model in which both datasets $X, Y \in \mathbb{R}^{d \times n}$ (or two point clouds of $n$ datapoints in $\mathbb{R}^d$) are sampled as follows. First, $X = (x_1, \ldots, x_n)$ is a collection of i.i.d. $\mathcal{N}(0, I_d)$ Gaussian vectors, and $Y = (y_1, \ldots, y_n)$ is a noisy version of $X = (x_1, \ldots, x_n)$, up to an orthogonal transformation $Q^\star$ and a relabeling $\pi^\star : [n] \to [n]$ of the data points, that is:

$$\forall i \in [n], \quad y_i = Q^\star x_{\pi^\star(i)} + \sigma z_i, \tag{1}$$

or, in matrix form:

$$Y = Q^\star X (P^\star)^\top + \sigma Z,$$

where $Z = (z_1, \ldots, z_n) \in \mathbb{R}^{d \times n}$ is also made of i.i.d. $\mathcal{N}(0, I_d)$ Gaussian vectors, $P^\star$ is the permutation matrix associated with some permutation $\pi^\star$, and $\sigma > 0$ is the noise parameter. Recovering (in some sense that will be made precise in the sequel) the (unknown) permutation $\pi^\star$ and orthogonal transformation $Q^\star$ defines the *Procrustes-Wasserstein problem* (sometimes abbreviated as PW in the sequel), which will be the focus of this study.

The practical approaches previously mentioned have shown good empirical results and are often scalable to large datasets. However, they suffer from a lack of theoretical results to guarantee their performance or to exhibit regimes where they fail. Model (1) described here above appears to be the simplest one to obtain such guarantees. We are interested in pinning down the *fundamental* limits of the Procrustes-Wasserstein problem, hence providing an ideal baseline for any computational method to be compared to, before delving into computational aspects. Our contributions are as follows:

$(i)$ We define a planted model for the Procrustes-Wassertein problem and discuss the appropriate choice of metrics to measure the performance of any estimator. Based on these metrics, we establish[1]:

$(i.a)$ information-theoretic results in the high-dimensional $d \gg \log n$ regime which was not explored before for this problem;

$(i.b)$ new information-theoretic results in the low-dimensional regime ($d \ll \log n$) for our metric of performance (the $L^2$ transport cost), which substantially differ from those obtained in Wang et al. [2022] for the overlap.

$(ii)$ We study computational aspects and propose the 'Ping-Pong algorithm', alternatively estimating the orthogonal transformation and the relabeling, initialized via a Franke-Wolfe convex relaxation. This method is quite close to that proposed in Grave et al. [2019] although the alternating part differs. We give sufficient conditions for the method to retrieve the planted signal after one single step.

$(iii)$ Finally, we provide experimental results to compare the proposed approach with the state-of-the-art method of Grave et al. [2019].

## 1.1 Discussion and related work

One can check that under the above model (1), the maximum likelihood (ML) estimators of $(P^\star, Q^\star)$ given $(X, Y)$ is given by:

$$(\hat{P}, \hat{Q}) \in \underset{(P,Q) \in \mathcal{S}_n \times \mathcal{O}(d)}{\arg\min} \frac{1}{n} \|XP^\top - Q^\top Y\|_F^2 = \underset{(P,Q) \in \mathcal{S}_n \times \mathcal{O}(d)}{\arg\min} \|QX - YP\|_F^2, \tag{2}$$

which is strictly equivalent[2] to the formulation of the non-planted problem of Grave et al. [2019]. Exactly solving the joint optimization problem (2) is non convex and difficult in general. However, if $P^\star$ is known then (2) boils down to the following *orthogonal Procrustes problem*:

$$\hat{Q} \in \underset{Q \in \mathcal{O}(d)}{\arg\min} \frac{1}{n} \|XP^\star - Q^\top Y\|_F^2, \tag{3}$$

which has a simple closed form solution given by $\hat{Q} = UV^\top$ where $USV^\top$ is the singular value decomposition (SVD) of $Y(XP^\star)^\top$ (see Schönemann [1966]). Conversely, when $Q^\star$ is known, (2) amounts to the following *linear assignment problem* (LAP in the sequel):

$$\underset{P \in \mathcal{S}_n}{\arg\min} \frac{1}{n} \|XP^\top - Q^\star Y\|_F^2 = \underset{P \in \mathcal{S}_n}{\arg\max} \frac{1}{n} \langle XP^\top, Q^\star Y \rangle, \tag{4}$$

---

[1]this very dichotomy ($d \gg \log n$ versus $d \ll \log n$) is fundamental in high dimensional statistics and not a mere artifact of our rationale; see Remark 1.

[2]their matrices $X$ and $Y$ are the transposed versions of ours.

| Reference | Setting | Metrics | Regime | Condition |
|---|---|---|---|---|
| Kunisky and Niles-Weed [2022] | $Q^\star = I_d$ | ov for $P^\star$ | $d \ll \log n$ | $\sigma \ll n^{-2/d}$ for $\mathrm{ov}(\hat{\pi}, \pi^\star) = 0$ 
 $\sigma \ll n^{-1/d}$ for $\mathrm{ov}(\hat{\pi}, \pi^\star) = o(1)$ |
| | | | $d \sim a \log n$ | $\sigma < (e^{4/a} - 1)^{-1/2}$ for $\mathrm{ov}(\hat{\pi}, \pi^\star) = 0$ 
 $\sigma < ((2e^{1/a} - 1)^2 - 1)^{-1/2}$ for $\mathrm{ov}(\hat{\pi}, \pi^\star) = o(1)$ |
| | | | $d \gg \log n$ | $\sigma < (1/2 - \varepsilon)(d/\log n)^{1/2}$ for $\mathrm{ov}(\hat{\pi}, \pi^\star) = 0$ |
| Wang et al. [2022] | $A = X^\top X, B = Y^\top Y$ | ov for $P^\star$ | $d \ll \log n$ | $\sigma \ll n^{-2/d}$ for $\mathrm{ov}(\hat{\pi}, \pi^\star) = 0$ 
 $\sigma \ll n^{-1/d}$ for $\mathrm{ov}(\hat{\pi}, \pi^\star) = o(1)$ |
| This paper | $X, Y$ from (1) | $c^2$ for $P^\star$, 
 $\ell^2$ for $Q^\star$ | $d \ll \log n$ 
 $d \gg \log n$ | $\sigma \ll d^{-1/2}$ for $c^2(\hat{\pi}, \pi^\star) = \ell^2(\hat{Q}, Q^\star) = o(d)$ 
 $\sigma \ll 1$ for $c^2(\hat{\pi}, \pi^\star) = \ell^2(\hat{Q}, Q^\star) = o(d)$ |

Table 1: Summary of previous informational results, together with the ones in this paper

which can be solved in polynomial time, e.g. in cubic time by the celebrated Hungarian algorithm [Kuhn, 1955], or more efficiently at the price of regularizing the objective and using the celebrated Sinkhorn algorithm [Cuturi, 2013].

*Previous results when $Q^\star$ is known.* As seen above, when $Q^\star$ is known (assume e.g. $Q^\star = I_d$), the Procrustes-Wasserstein problem reduces to a simpler objective, that of aligning Gaussian databases. This problem has been studied by Dai et al. [2019, 2023] in the context of feature matching. Kunisky and Niles-Weed [2022] study the same problem as a geometric extension of planted matching and establish state-of-the-art statistical bounds in the Gaussian model in the low-dimensional ($d \ll \log n$), logarithmic ($d \sim a \log n$) and high-dimensional ($d \gg \log n$) regimes. In particular, they show that exact recovery is feasible in the logarithmic regime $d \sim a \log n$ if $\sigma^2 < \frac{1}{e^{4/a}-1}$, and in the high-dimensional regime if $\sigma^2 < (1/4 - \varepsilon)\frac{d}{\log n}$. Note that in this problem, there is no computational/statistical gap since the LAP is always solvable in polynomial time.

*Geometric graph alignment.* Strongly connected to the Procrustes-Wassertein problem is the topic of graph alignment where the instances come from a geometric model. Wang et al. [2022] investigate this problem for complete weighted graphs. In their setting, given a permutation $\pi^\star$ on $[n]$ and $n$ i.i.d. pairs of correlated Gaussian vectors $(X_{\pi^\star(i)}, Y_i)$ in $\mathbb{R}^d$ with noise parameter $\sigma$, they observe matrices $A = X^\top X$ and $B = Y^\top Y$ (i.e all inner products $\langle X_i, X_j \rangle$ and $\langle Y_i, Y_j \rangle$) and are interested in recovering the hidden vertex correspondence $\pi^\star$. The maximum likelihood estimator in this setting writes

$$\arg\min_{P \in \mathcal{S}_n} \frac{1}{n} \|X^\top X P - P Y^\top Y\|_F^2 = \arg\max_{P \in \mathcal{S}_n} \frac{1}{n} \langle P^\top X^\top X P, Y^\top Y \rangle, \tag{5}$$

which is an instance of the *quadratic assignment problem* (QAP in the sequel), known to be NP-hard in general, as well as some of its approximations [Makarychev et al., 2014]. In fact, we have the following informal equivalence (see Appendix A for a proof):

**Lemma 1** (Informal). *PW and geometric graph alignement are equivalent, that is, one knows how to (approximately) solve the former iff they know how to (approximately) solve the latter.*

Wang et al. [2022] focus on the low-dimensional regime $d = o(\log n)$, where geometry plays the most important role (see Remark 1). They prove that exact (resp. almost exact) recovery of $\pi^\star$ is information-theoretically possible as soon as $\sigma = o(n^{-2/d})$ (resp. $\sigma = o(n^{-1/d})$). They conduct numerical experiments which suggest good performance of the celebrated Umeyama algorithm [Umeyama, 1988], which is confirmed by a follow-up work by Gong and Li [2024] analyzing the Umeyama algorithm (which is polynomial time in the low dimensional regime $d = o(\log n)$) in the same setting and shows that it achieves exact (resp. almost exact) recovery of $\pi^\star$ if $\sigma = o(d^{-3}n^{-2/d})$ (resp. $\sigma = o(d^{-3}n^{-1/d})$), hence coinciding with the information thresholds up to a $\mathrm{poly}(d)$ factor. However, their algorithm is of time complexity at least $\Omega(2^d n^3)$, which is not polynomial in $d$. This is why we do not include this method in our baselines.

We emphasize that our results clearly depart from those obtained in Wang et al. [2022] and Gong and Li [2024], because $(i)$ we are also interested in the high dimensional case $d \gg \log n$, and $(ii)$ we work with a different performance metric which provides less stringent conditions for the recovery to be feasible, see Section 1.2. A summary of previous informational results together with ours (see also Section 2) is given in Table 1.

*On the orthogonal transformation $Q^\star$.* Generalizing the standard linear assignment problem, our model described above in (1) introduces an additional orthogonal transformation $Q^\star$ across the datasets. This orthogonal transformation can be motivated in the context of aligning embeddings in a

high-dimensional space: indeed, the task of learning embeddings is often agnostic to orientation in the latent space. In other words, two point clouds may represent the same data points while having different global orientations. Hence, across different data sets, learning this orientation shift is crucial in order to compare (or align) the point clouds. As an illustration of this fact, Xing et al. [2015] provides empirical evidence that orthogonal transformations are particularly adapted for bilingual word translation.

*Discussing the method proposed in Grave et al. [2019].* We conclude this introduction by discussing the work of Grave et al. [2019]. Their proposed algorithm is as follows. At each iteration $t$, given a current estimate $Q_t$ of the orthogonal transformation, we sample mini-batches $X_t, Y_t$ of same size $b$ and find the optimal matching $P_t$ between $Y_t Q_t^\top$ and $X_t$, via solving a linear assignment problem of size $b$. This matching $P_t$ in turn helps to refine the estimation of the orthogonal transformation via a projected gradient descent step, and the procedure repeats. This method has the main advantage to be scalable to very large datasets and to perform well in practice ; however, no guarantees are given for this method, and in particular the mini-batch step which can justifiably raise some concerns. Indeed, since $X_t = (x_{t,j})_{j \in [b]}$ and $Y_t = (y_{t,j})_{j \in [b]}$ are chosen independently, if $b \ll \sqrt{n}$ it is likely that for any matching $\pi_t$ the pairs $(x_{t,j}, y_{t, \pi_t(j)})$ always correspond to disjoint pairs, and thus aligning $Y_t Q_t^\top$ and $X_t$ does not reveal any useful information about the true $P^\star$ – this is even more striking when the data is non-isotropic.

## 1.2  Problem setting and metrics of performance

*Notations.* We denote by $X \sim \mathcal{N}(\mu, \Sigma)$ with $\mu \in \mathbb{R}^d$ and $\Sigma \in \mathbb{R}^{d \times d}$ the fact that $X$ follows a Gaussian distribution in $\mathbb{R}^d$ of mean $\mu$ and covariance matrix $\Sigma$. If $\mu = 0$ and $\Sigma = I_d$, variable $X$ is called *standard Gaussian*. We denote by $\mathcal{O}(d)$ the orthogonal group in dimension $d$, and by $\mathcal{S}_n$ the group of permutations on $[n]$. Throughout, $\| \cdot \|$ and $\langle \cdot \rangle$ are is the standard euclidean norm and scalar product on $\mathbb{R}^d$, and $\| \cdot \|_F$ and $\| \cdot \|_{op}$ are respectively the Frobenius matrix norm and the operator matrix norm. The spectral radius of a matrix $A$ is denoted $\rho(A)$. In all the proofs, quantities $c_i$ where $i$ is an integer are unspecified constants which are universal, that is independent from the parameters. Finally, all considered asymptotics are when $n \to \infty$. Note that $d$ also depends on $n$. An event is said to hold *with high probability (w.h.p.)* if its probability tends to 1 when $n$ goes to $\infty$.

*Problem setting and performance metrics.* We work with the planted model as introduced in (1) and recall that our goal is to recover the permutation $\pi^\star$ and the orthogonal matrix $Q^\star$ from the observation of $X$ and $Y$.

*Performance metrics.* Previous works measure the performance of an estimator $\hat{\pi}$ of a planted relabeling $\pi^\star$ via the overlap:

$$\mathrm{ov}(\pi, \pi') := \frac{1}{n} \sum_{i=1}^{n} \mathbf{1}_{\{\hat{\pi}(i) = \pi'(i)\}}, \tag{6}$$

defined for any two permutations $\pi, \pi'$. This is an interesting metric when we have no hierarchy in the errors, that is when only the true match is valuable, and all wrong matches cost the same. However, this discrete measure does not take into account the underlying geometry of the model. A performance metric which is more adapted to our setting is the $L^2$ transport cost between the point clouds. The natural intuition is that a mismatch is less costy if it corresponds to embeddings which are in fact close in the underlying space. We define

$$c^2(\pi, \pi') = \frac{1}{n} \sum_{i=1}^{n} \left\| x_{\pi(i)} - x_{\pi'(i)} \right\|^2,$$

for any two permutations $\pi, \pi'$. Note that this cost can also be written in matrix form as $c^2(P, P') = \left\| (P - P') X^\top \right\|_F^2$. From this form it is clear that, as stated before, $c^2(P, P')$ is nothing but the euclidean transport cost for aligning $XP^\top$ onto $X(P')^\top$. Note that these two measures, ov and $c^2$, are also well-defined[3] when $P, P'$ are more general (and in particular when they are bistochastic matrices). Finally, we measure the performance for the estimation of $Q^\star$ via the Frobenius norm:

$$\ell^2(Q, Q') = \|Q - Q'\|_F^2,$$

---

[3]for the overlap, one could extend its definition using that $\mathrm{ov}(P, P') = \langle P, P' \rangle$.

defined for any two orthogonal matrices $Q, Q'$.

*Comparison between metrics.* For a Haar-distributed matrix $Q$ on $\mathcal{O}(d)$, we have that $\mathbb{E}\left[\ell^2(Q, Q^\star)\right] = 2d$, while for $\pi$ sampled uniformly from the set of all permutations, we have $\mathbb{E}\left[c^2(\hat{\pi}, \pi^\star)\right] = 2d(1 - 1/n)$ and $\mathbb{E}\left[\mathrm{ov}(\hat{\pi}, \pi^\star)\right] = 1/n$. Hence, some estimators $\hat{\pi}, \hat{Q}$ of $\pi^\star, Q^\star$ will perform well in our metrics if they can achieve $\ell^2(Q, Q^\star) \leqslant \varepsilon d$, and $c^2(\pi, \pi^\star) \leqslant \varepsilon d$ for some small (possibly vanishing) $\varepsilon > 0$.

Depending on dimension $d$, similarity measures given by $c^2$ and the overlap can behave differently or coincide. In the case where $d$ is small, and thus plays a very important role, $\mathrm{ov}$ and $c^2$ have very different behaviors, and lead to very different results. In particular, there is a wide regime in which inferring $\pi^\star$ for the overlap sense is impossible, but reachable in the transport cost sense, see Section 2.

For any fixed permutation $\pi$, we have that $\mathbb{E}\left[c^2(\pi, \pi^\star)\right] = 2d(1 - \mathrm{ov}(\pi, \pi^\star))$, where the mean is taken with respect to the randomness of $X$. We also have the basic deterministic inequality

$$c^2(\pi, \pi^\star) \leqslant (1 - \mathrm{ov}(\pi, \pi^\star)) \times \sup_{(i,j) \in [n]^2} \|x_i - x_j\|^2 .$$

Thus, as long as $\sup_{(i,j) \in [n]^2} \|x_i - x_j\|^2 = O(d)$, an estimator $\hat{\pi}$ with good overlap $(1 - \mathrm{ov}(\pi, \pi^\star) \leqslant \varepsilon)$ also has a good $c^2$ cost $(c^2(\pi, \pi^\star) = O(\varepsilon d))$. However, this required control $\sup_{(i,j) \in [n]^2} \|x_i - x_j\|^2 = O(d)$ only holds as long as $d \gg \log(n)$.

The blessing of large dimensions lead to an equivalence between the discrete metric $\mathrm{ov}$, and the continuous transport metric $c^2$. We gather several important points highlighting the dichotomy between small and large dimensions for our problem in the following remark.

**Remark 1.** *On the blessings of large dimensions for our problem:*

1. *For any open ball $\mathcal{B}$ of radius $\varepsilon > 0$, denoting $\mathcal{X} = \{x_i, i \in [n]\}$, we have that $\mathbb{P}(\mathcal{B} \cap \mathcal{X} = \varnothing) \to 1$ if $d \gg \log(n)$, while if $d \ll \log(n)$ then for all $M > 0$, $\mathbb{P}(|\mathcal{B} \cap \mathcal{X}| \geqslant M) \to 1$. In small dimensions, any fixed non-empty ball will contain infinitely many points of $\mathcal{X}$ as $n$ increases, while in large dimensions these points are separated and any fixed ball will contain no such points w.h.p.*

2. *For $d \gg \log(n)$, matrix $X/\sqrt{d}$ satisfies the* restricted isometry property *[Candès, 2008].*

3. *For $d \gg \log(n)$, the overlap and the transport cost metrics are equivalent: there exist numerical constants $\alpha, \beta > 0$ such that w.h.p., for all permutation matrices $\pi, \pi'$, $\alpha c^2(\pi, \pi') \leqslant 2d(1 - \mathrm{ov}(\pi, \pi')) \leqslant \beta c^2(\pi, \pi')$.*

*Organization of the rest of the paper.* Section 2 is dedicated to our informational results, giving their essential content as well as the main ideas on the proofs. We next discuss in Section 3 some computational results, introducing the Ping-Pong algorithm, and presenting our numerical experiments.

## 2  Informational results

The substantial theoretic part of the paper stands in the informational results obtained for the Procrustes-Wasserstein problem which we describe hereafter.

### 2.1  High dimensions

In the high-dimensional case when $\log n \ll d$ (and $d \log d \ll n$), our results – Theorem 1 below – imply that if $\sigma \to 0$ then the ML estimators defined in (2) satisfy w.h.p.

$$\mathrm{ov}(\pi^\star, \hat{\pi}) = 1 - o(1), \ c^2(\hat{P}, P^\star) = o(d), \text{ and } \ell^2(\hat{Q}, Q^\star) = o(d),$$

that is one can infer $\pi^\star$ and $Q^\star$ almost exactly, for all introduced metrics, as soon as $\sigma \to 0$.

Note that this is the first result in the high-dimensional regime for the Procrustes Wassertein problem: Kunisky and Niles-Weed [2022] also considered this regime but only for the LAP problem (that

is recovering $\pi^\star$ when $Q^\star$ in known), and the only existing results for geometric graph alignment Wang et al. [2022], Gong and Li [2024] do not consider this high dimensional case. Our result thus complements the existing picture and shows that almost exact recovery is feasible under the loose assumption $\sigma \to 0$, in the $c^2$ and the overlap sense, since these metrics are equivalent in large dimensions (see Remark 1). Our result is in fact more specific and only requires $d \geqslant 2 \log n$. We prove the following Theorem:

**Theorem 1.** *Assume that $d \geqslant 2 \log n$. There exists universal constants $c_1, c_2, c_3 > 0$ so that for $n$ large enough, with probability $1 - o(1)$, the ML estimators defined in* (2) *satisfy*

$$\mathrm{ov}(\pi^\star, \hat{\pi}) \geqslant 1 - \max\left(60\sigma^2, c_1 \frac{d}{n}, c_2 \frac{\log n}{d \log d}\right), \tag{7}$$

*and*

$$\frac{\ell^2(Q^\star, \hat{Q})}{2d} \leqslant c_1 \frac{d}{n} + c_2 \sigma^2 + c_3 \max\left(\frac{d \log n}{n}, \sqrt{\frac{\log n}{n}}\right). \tag{8}$$

The proof of Theorem 1 is detailed in Appendix C and builds upon controlling the probability of existence of a certain subset of indices $\mathcal{K}(\hat{Q}, \hat{\pi}, Q^\star)$ of vectors with prescribed properties in order to show that $\pi^\star$ can be recovered. We apply standard concentration inequalities to control the previous probability. The $d \geqslant 2 \log(n)$ assumption is crucial here since it allows the union bound over $\mathcal{S}_n$ to work.

## 2.2 Low dimensions

In the low-dimensional case when $d \ll \log n$, Theorem 2 below implies that if $\sigma = o(d^{-1/2})$ then there exist estimators $\hat{\pi}, \hat{Q}$ that satisfy w.h.p.

$$c^2(\hat{P}, P^\star) = o(d), \text{ and } \ell^2(\hat{Q}, Q^\star) = o(d),$$

that is, one can approximate $\pi^\star$ (in the $c^2$ sense *only*) and $Q^\star$ as soon as $\sigma = o(d^{-1/2})$. This is of course to be put in contrast with the previous results on geometric graph alignment in this low-dimensional regime: for almost exact recovery in Wang et al. [2022] *in the overlap sense*, we need $\sigma = o(n^{-1/d})$, which is far more restrictive than $\sigma = o(d^{-1/2})$ as soon as $d \log(d) < \log n$, that is nearly in the whole low dimensional regime when $d \ll \log n$. In particular, since the rates of Wang et al. [2022] are sharp when $d$ is of constant order, in order to approximate $\pi^\star$ in the overlap sense it is necessary to have $\sigma$ to decreasing polynomially (at rate $1/n^{1/d}$) to 0, whereas approximating $\pi^\star$ in the transport cost sense requires only $\sigma = o(1)$.

There is no contradiction here, since we recall that the $c^2$ metric and the overlap are not equivalent in small dimensions: let us give a few more insights on this. This scaling $n^{-1/d}$ comes from the fact that in small dimensions, points of the dataset are close to each other, and the order of magnitude between some $x_i$ and its closest point in the dataset scales exactly as $n^{-1/d}$: if the noise is smaller than this quantity, one should be able to recover the planted permutation. However, when it comes to considering the $c^2$ metric, matching $i$ with $j$ such that $\|x_i - x_j\|^2 \ll d$ is sufficient, thus suggesting that recovering a permutation with small $c^2$ cost and recovering $Q^\star$ with small Frobenius norm error should be achievable even with large $\sigma$ (*i.e.*, that does no tend to 0 as $n$ increases).

Our main theorem for low dimensions is as follows.

**Theorem 2.** *Let $\delta_0 \in (0, 1)$. There exist estimators $\hat{\pi}, \hat{Q}$ of $\pi^\star, Q^\star$ such that if for some numerical constants $C_1, C_2 > 0$ we have $\sigma \leqslant C_1 \delta_0^2 d^{-1/2}$ and $\log(n) \geqslant C_2 d \log(1/\delta_0)$, then:*

$$\frac{c^2(\hat{\pi}, \pi^\star)}{2d} \leqslant \delta_0 \quad and \quad \frac{\ell^2(\hat{Q}, Q^\star)}{2d} \leqslant \delta_0.$$

A refined version of Theorem 2, namely Theorem 3, is proved in Appendix D. We emphasize that the estimators considered in Theorem 2 are *not* the ML estimators: recall that the strategy to analyse the former as rolled out for Theorem 1 required the union bound over $\mathcal{S}_n$ to work. This drastically fails when $d \ll \log(n)$. Hence, we will instead focus on an estimator that takes advantage of the fact that $d$ is small, and show that even in small dimensions, the signal-to-noise ratio $\sigma$ does not need to decrease with $n$.

Let us first describe the intuition behind the estimators $\hat{\pi}, \hat{Q}$. When $d = 1$, $Q^{\star} = \pm 1$ and a simple strategy to recover $Q^{\star}$ is to count the number $N_+(\mathcal{X}), N_-(\mathcal{X})$ (resp. $N_+(\mathcal{Y}), N_-(\mathcal{Y})$) of positive and negative $x_i$ (resp. positive and negative $y_j$): if $N_+(\mathcal{X})$ and $N_+(\mathcal{Y})$ are close, then we output $\hat{Q} = +1$, whereas if $N_+(\mathcal{X})$ and $N_-(\mathcal{Y})$ are close, then $\hat{Q} = -1$. In dimension $d$, an analog strategy can be applied at the cost of looking in all relevant directions, and the number of such directions is exponentially big in $d$. Our strategy is thus as follows. We compute the number of points that lie in a given cone $\mathcal{C}(u, \delta)$ of given angle $\delta$ and direction $u$. Then, we estimate $Q^{\star}$ by the orthogonal transformation $\hat{Q}$ which makes the number of $y_j$ in $\mathcal{C}(u, \delta)$ closest to the number of $x_j$ in $\mathcal{C}(\hat{Q}u, \delta)$, for any direction $u$. Note that this approach heavily relies on the small dimension assumption $d \ll \log n$: in this case, for any constant $\delta$, all theses cones contain w.h.p. a large number of points (tending to $\infty$ with $n$), which does not hold anymore when $d \gg \log n$.

For $\delta > 0$ and $u \in \mathcal{S}^{d-1}$, let $\mathcal{C}(u, \delta) := \left\{ v \in \mathbb{R}^d \,|\, \langle u, v \rangle \geqslant (1-\delta)\|v\| \right\}$ be the cone of angle $\delta$ centered around $u$. Let $\mathcal{X} := \{x_i, i \in [n]\}$, $\mathcal{Y} := \{y_i, i \in [n]\}$. We now introduce the following sets, for some $\kappa > 0$:

$$\mathcal{C}_{\mathcal{X}}(u, \delta) := \mathcal{X} \cap \mathcal{C}(u, \delta) \cap \mathcal{B}(0, 1/\kappa)^C \quad \text{and} \quad \mathcal{C}_{\mathcal{Y}}(u, \delta) := \mathcal{Y} \cap \mathcal{C}(u, \delta) \cap \mathcal{B}(0, \sqrt{1+\sigma^2}/\kappa)^C,$$

where $\mathcal{B}(0, r)^C$ contains all vectors in $\mathbb{R}^d$ of norm larger than or equal to $r$. The role of $\kappa > 0$ is to prevent side effects: indeed, since the cones are centered at the origin, points that are too close to $0$ fall into cones with arbitrary directions and are not informative for the statistics we want to compute.

Now, for some $p \geqslant 1$ and directions $u_1, \ldots, u_p \in \mathcal{S}^{d-1}$ to be set later, we define the following *conical alignment loss*:

$$\forall Q \in \mathcal{O}(d), \quad F(Q) = \frac{1}{p} \sum_{k=1}^{p} \left( |\mathcal{C}_{\mathcal{X}}(Qu_k, \delta)| - |\mathcal{C}_{\mathcal{Y}}(u_k, \delta)| \right)^2. \tag{9}$$

The estimator $\hat{Q}$ in Theorem 2 is then defined as a minimizer of the conical alignment loss over a finite set $\mathcal{N} \subseteq \mathcal{O}(d)$:

$$\hat{Q} \in \underset{Q \in \mathcal{N}}{\arg\min} \, F(Q),$$

where $\mathcal{N}$ will further be some $\varepsilon$-net of $\mathcal{O}(d)$, while $\hat{\pi}$ is then obtained by a LAP as in (10).

## 2.3 From $P^{\star}$ to $Q^{\star}$ and vice versa

In our proofs, we often prove that one of the estimators $\hat{P}$ or $\hat{Q}$ performs well in order to deduce that both perform well. This is thanks to the following two results, proved in Appendix B.1 and B.2.

**Lemma 2** (From $\hat{Q}$ to $\hat{P}$). *Let $\delta \in (0, 1/2)$. Assume that there exists $\hat{Q}$ that is $\sigma(\{x_1, \ldots, x_n, y_1, \ldots, y_n\})$-measurable such that $\ell_{\mathrm{ortho}}^2(\hat{Q}, Q^*) := \|\hat{Q} - Q^*\|^2 \leqslant \delta d$. There exist constants $C_1, C_2, C_3 > 0$ such that with probability at least $1 - 2e^{-nd} - 2e^{-(d^2+\sqrt{n})}$,*

$$\hat{\pi} \in \arg\min_{\pi \in \mathcal{S}_n} \frac{1}{n} \sum_{i=1}^{n} \left\| x_{\pi(i)} - \hat{Q}^{\top} y_i \right\|^2, \tag{10}$$

*that can be computed in polynomial time (complexity $O(n^3)$) as the solution of a LAP, satisfies:*

$$\frac{c^2(\hat{\pi}, \pi^{\star})}{d} \leqslant C_1 \delta + C_2 \sigma^2 + C_3 \max\left( \frac{d \ln(1/\delta)}{n}, \sqrt{\frac{\ln(1/\delta)}{n}} \right).$$

**Lemma 3** (From $\hat{P}$ to $\hat{Q}$). *Let $\delta \in (0, 1/2)$. Assume that there exists $\hat{\pi}$ that is $\sigma(\{x_1, \ldots, x_n, y_1, \ldots, y_n\})$-measurable such that $c^2(\hat{\pi}, \pi^{\star}) \leqslant \delta d$. Let $\hat{Q}$ be the solution to the following optimization problem: There exist constants $C_1, C_2, C_3 > 0$ such that with probability at least $1 - 2e^{-nd} - 2e^{-(d^2+\sqrt{n})}$,*

$$\hat{Q} \in \arg\min_{Q \in \mathcal{O}(d)} \frac{1}{n} \sum_{i=1}^{n} \left\| x_{\hat{\pi}(i)} - Q^{\top} y_i \right\|^2, \tag{11}$$

*that can be computed in closed form with an SVD of $XY^{\top}$, satisfies:*

$$\frac{\ell_{\mathrm{ortho}}^2(\hat{Q}, Q^{\star})}{d} \leqslant C_1 \delta + C_2 \sigma^2 + C_3 \max\left( \frac{d \ln(1/\delta)}{n}, \sqrt{\frac{\ln(1/\delta)}{n}} \right).$$

## 3 Computational aspects

The estimators provided this far in Section 2, namely the joint minimization in $P$ and $Q$ in (2) and the minimizer of the conical alignment loss in (9) are of course not poly-time in general. In this section, we are interested in computational aspects of the problem.

### 3.1 Convex relaxation and Ping-Pong algorithm

Estimating $P^\star$ can be made via solving the QAP (5), that can be convexified into the *relaxed quadratic assignment problem* (relaxed QAP):

$$\hat{P}_{\text{relaxed}} \in \arg\min_{P \in \mathcal{D}_n} \frac{1}{n} \|X^\top X P - P Y^\top Y\|_F^2 \,, \tag{12}$$

where $\mathcal{D}_n$ is the polytope of *bistochastic* matrices, which is the convex envelope of the set of permutation matrices. Note that unlike in (5), this argmin is not necessarily equal to $\arg\max_{P \in \mathcal{D}_n} \langle P^\top X^\top X P, Y^T Y' \rangle$ since $\mathcal{D}_n$ contains non-orthogonal matrices.

The estimate $\hat{P}_{\text{relaxed}}$ gives a first estimate to then perform alternate minimizations in $Q$ through an SVD – see (11) – and $P$ through a LAP – see (10). Combining an initialization with convex relaxation, computed via Frank-Wolfe algorithm [Jaggi, 2013] and the alternate minimizations in $P$ and $Q$ yields the *Ping-Pong algorithm*.

---

**Algorithm 1:** PING-PONG ALGORITHM

**Input:** Number of Frank-Wolfe steps $T$, number of alternate-minimization steps $K$, $\tilde{P}_0 = \frac{\mathbf{1}\mathbf{1}^\top}{n}$

1 **for** $k = 0$ *to* $T - 1$ **do**
2 $\quad$ Compute $S_k = \arg\min_{P \in \mathcal{S}_n} \langle P, \nabla f(\tilde{P}_k) \rangle$ (LAP), where $f(P) = \left\| X^\top X P - P Y^\top Y \right\|_F^2$
3 $\quad$ $\tilde{P}_{k+1} = (1 - \gamma_k)\tilde{P}_k + \gamma_k S_k$ for $\gamma_k = \frac{1}{2+k}$
4 $P_0 = \tilde{P}_T$ and $Q_0 = I_d$
5 **for** $k = 0$ *to* $K - 1$ **do**
6 $\quad$ $Q_{k+1} = U_k V_k^\top$ for $Y P_k X^\top = U_k D_k V_k$ the SVD of $Y P_k X^\top$ $\qquad$ *(Ping)*
7 $\quad$ $P_{k+1} \in \arg\max_{P \in \mathcal{S}_n} \langle P, Y^\top Q_{k+1} X \rangle$ (LAP) $\qquad$ *(Pong)*
**Output:** $P_K, Q_K$

---

Algorithm 1 is structurally similar to Grave et al. [2019]'s algorithm, as explained in the introduction. The difference lies in the steps in Lines 6-7 of Algorithm 1: while Grave et al. [2019] perform projected gradient steps, our approach is more greedy and directly minimizes in each variable. Both approaches are experimentally compared in Section 3.3.

### 3.2 Guarantees for one step of Ping-Pong algorithm

Providing statistical rates for the outputs of Algorithm 1 is a challenging problem for two reasons. First, relaxed QAP is not a well-understood problem: the only existing guarantees in the literature are for correlated Gaussian Wigner models in the noiseless case (i.e., $\sigma = 0$ in our model) [Valdivia and Tyagi, 2023], while for correlated Erdös-Rényi graphs, the relaxation is known to behave badly in general [Lyzinski et al., 2016]. Secondly, studying the iterates in lines 6 and 7 of the algorithm is challenging, since these are projections on non-convex sets. While Lemmas 2 and 3 show that if $P_k$ (resp. $Q_k$) has small $c^2$ loss, then $Q_{k+1}$ has small $\ell^2$ loss (resp. $P_{k+1}$ has small $c^2$ loss), showing that there is a contraction at each iteration 'à la Picard's fix-point Theorem' remains out of reach for this paper. We thus resort to proving that *one single step* of Algorithm 1 ($K = T = 1$) can recover the planted signal, provided that the noise $\sigma$ is small enough.

**Proposition 1.** *There exists $C > 0$ such that for any $\delta \in (0, 1)$, if $\sigma \leqslant n^{-\frac{13}{\delta}}$, then the permutation $\hat{\pi}$ associated to the outputs $\hat{\pi}, \hat{Q}$ of Algorithm 1 for $K = T = 1$ satisfies, with probability $1 - 1/n$:*

$$\text{ov}(\pi^\star, \hat{\pi}) \geqslant 1 - \delta \,.$$

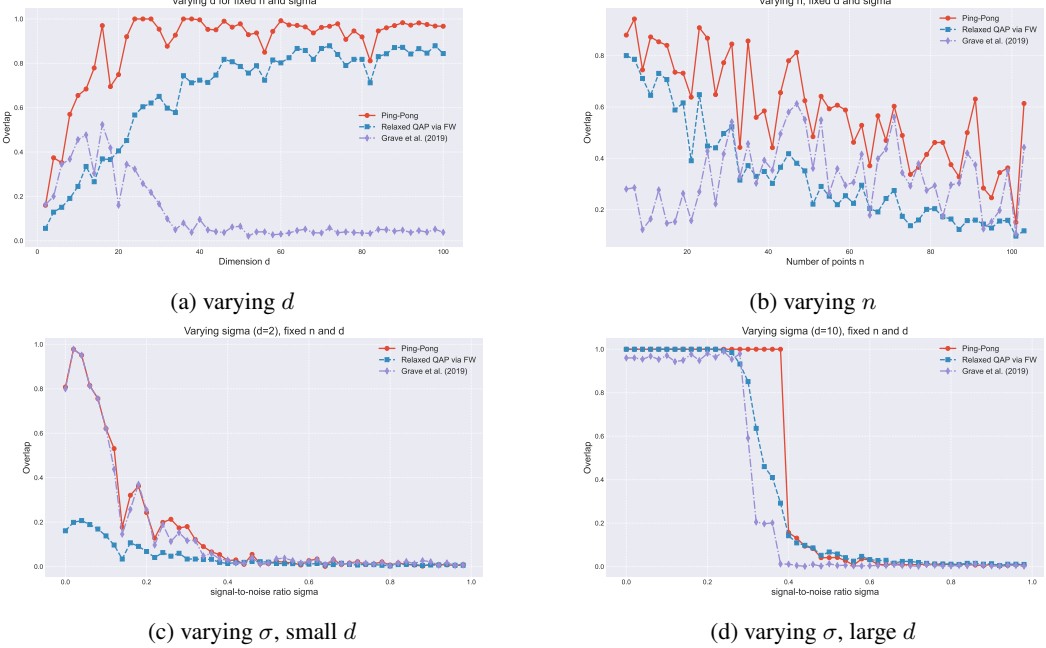

(a) varying $d$              (b) varying $n$

(c) varying $\sigma$, small $d$         (d) varying $\sigma$, large $d$

Figure 1: Influence of the parameters (dimensions $d$, number of points $n$, and noise level $\sigma$) on the accuracy (in terms of overlap) of three different estimators: the relaxed QAP estimator (12) projected on the set of permutation matrices (blue curve), the output of Alg. 1 (red curve), and the output of Grave et al. [2019]'s algorithm (purple curve). Each dot corresponds to averaging scores over 10 experiments. Figure 1a: $\sigma = 0.34, n = 100$. Figure 1b: $\sigma = 0.34, d = 5$. Figures 1c and 1d: $n = 200, d = 2$ and $d = 60$ respectively.

*In the high-dimensional setting ($d \gg \log(n)$), there exist some constants $c_1, c_2$ such that if $\sigma \leqslant n^{-c_1}$, then $\hat{\pi}$ satisfies w.h.p.*

$$\mathrm{ov}(\pi^\star, \hat{\pi}) \geqslant 1 - c_2 \max\left( \sqrt{\frac{d\log(d)}{n} + \frac{\log(n)}{d}}, \frac{d\log(d)}{n} + \frac{\log(n)}{d} \right).$$

Thus, for $\sigma$ polynomially small in $n$ and exponentially small in $1/\delta$, one step of Alg. 1 recovers $\pi^\star$ in the overlap sense with error $\delta$. In large dimensions, this is improved, since $\sigma$ is no longer required to be exponentially small as the target error decreases to zero. Proof of Proposition 3 is given in Appendix E.

### 3.3 Numerical experiments

We compare in Figure 1 our Alg. 1 with $(i)$ the naive initialization of the relaxed QAP estimator (12), and $(ii)$ the method in Grave et al. [2019]. The curve 'relaxed QAP via FW' is obtained by computing the relaxed QAP estimator with Frank-Wolfe algorithm with $T = 1000$ steps, enough for convergence. This estimator is then taken as initialization for Alg. 1 and Grave et al. [2019]'s algorithm, that are both taken with the same large number of steps ($K = 100$, empirically leading to convergence to stationary points of the algorithms). For fair comparison, we take full batches in Grave et al. [2019] (smaller batches lead to even worse performances).

## Conclusion

We establish new informational results for the Procrustes-Wassertein problem, both in the high ($d \gg \log n$) and low ($d \ll \log n$) dimensional regimes. We propose the 'Ping-Pong algorithm', alternatively estimating the orthogonal transformation and the relabeling, initialized via a Franke-Wolfe convex relaxation. Our experimental results show that our method most globally outperforms the algorithm proposed in Grave et al. [2019].

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

# A Useful results

We start by proving Lemma 1 which gives the equivalence between PW and geometric graph alignement.

## A.1 Proof of Lemma 1

*Proof of Lemma 1.* We have by Lemma 3 that as soon as we are able to estimate $\pi^\star$ with a small error in PW, we are also capable of doing so $Q^\star$, by perfoming a simple Singular Value Decomposition (SVD). Since one can trivially form an instance $A = X^\top X$ and $B = Y^\top Y$ of geometric graph alignement from an instance $(X, Y)$ of PW from model (1), we can deduce that if we know how to (approximately) solve geometric graph alignement, we know how to (approximately) solve PW.

Conversely, if we are given adjacency matrices $A = X^\top X, B = Y^\top Y$ of two correlated random geometric graphs under the Gaussian model from [Wang et al., 2022, Gong and Li, 2024] where $y_i = x_{\pi^\star(i)} + \sigma z_i$, we can recover $\pi^\star$ via solving PW. Indeed, $A$ is of rank at most $d$, so we can build $X' = (x'_1 | \dots | x'_n) \in \mathbb{R}^{d \times n}$ such that $A = X'^\top X'$. Similarly, we can build $Y' = (y'_1 | \dots | y'_n) \in \mathbb{R}^{d \times n}$ such that $B = Y'^\top Y'$. We have $X^\top X = X'^\top X'$, hence $\langle x_i, x_j \rangle = \langle x'_i, x'_j \rangle$, thus there exists $Q_1 \in \mathcal{O}(d)$ such that for all $i \in [n]$, $x'_i = Q_1 x_i$. Similarly, there exists $Q_2 \in \mathcal{O}(d)$ such that for all $i$, $y'_i = Q_2 y_i$. By multiplying these two orthogonal matrices by independent random uniform orthogonal matrices, we can always assume that they are independent from $X$ and $Y$. We obtained $X', Y'$ which satisfy $y'_i = Q^\star x'_{\pi^\star(i)} + \sigma z'_i$ for all $i$, where $Q^\star = Q_2 Q_1^\top$, and $x'_i = Q_1 x_i, z'_i = Q_2 z_i$ are i.i.d. standard Gaussian vectors. This is exactly an instance of the PW problem. If we know how to (approximately) solve the PW problem, we know how to (approximately) recover $\pi^\star$ and thus (approximately) solve the geometric graph alignment problem.

This proves that PW and geometric graph alignment are equivalent. $\qquad\square$

## A.2 $\varepsilon-$nets of $\mathcal{O}(d)$

Throughout the proofs, we will need to give high probability bounds on quantities for all orthogonal matrices. This is done by covering $\mathcal{O}(d)$ by a finite number of open balls centered at points of $\mathcal{O}(d)$. This is done by considering $\varepsilon-$nets.

**Definition 1** ($\varepsilon-$nets of $\mathcal{O}(d)$)**.** *Let $\varepsilon > 0$. A subset $\mathcal{N}_\varepsilon \subseteq \mathcal{O}(d)$ is a $\varepsilon-$net of $\mathcal{O}(d)$ for the Frobenius norm if for all $O \in \mathcal{O}(d)$ there exists $O_\varepsilon \in \mathcal{N}_\varepsilon$ such that $\|O - O_\varepsilon\|_F \leqslant \varepsilon$.*

**Remark 2.** *Note that since $\| \cdot \|_F \leqslant \| \cdot \|_{op}$ by Cauchy-Schwarz any $\varepsilon-$net of $\mathcal{O}(d)$ for the Frobenius norm is also an $\varepsilon-$net of $\mathcal{O}(d)$ for the operator norm.*

We will need $\varepsilon-$nets of $\mathcal{O}(d)$ that are not too large, in order to apply union bounds which will give non-trivial probabilistic controls. Guarantees on such $\varepsilon-$nets are standard in the literature; we give one which will be useful for us in the following Lemma.

**Lemma 4** ($\varepsilon-$nets of $\mathcal{O}(d)$ of minimal size, see e.g. Rogers [1963])**.** *There exists a universal constant $C > 0$ such that for all $\varepsilon > 0$, there exists an $\varepsilon-$net $\mathcal{N}_\varepsilon$ of $\mathcal{O}(d)$ such that*

$$|\mathcal{N}_\varepsilon| \leqslant \left( \frac{C\sqrt{d}}{\varepsilon} \right)^{d^2} .$$

# B Remaning proofs of Section 2

## B.1 Proof of Lemma 2

*Proof of Lemma 2.* Denote $g(\pi) := \frac{1}{n} \sum_{i=1}^n \left\| x_{\pi(i)} - \hat{Q}^\top y_i \right\|^2$. The proof relies on noticing that for all $\pi \in \mathcal{S}_n$, by definition $g(\hat{\pi}) \leqslant g(\pi)$ and using $\|a + b\|^2 \geqslant \frac{1}{2}\|a\|^2 - \|b\|^2$, one gets

$$g(\hat{\pi}) \geqslant \frac{1}{2} c^2(\hat{\pi}, \pi) - g(\pi),$$

and thus $c^2(\hat{\pi}, \pi) \leqslant 4g(\pi)$. We apply the previous inequality to $\pi = \pi^\star$ and using $\|a + b\|^2 \leqslant 2(\|a\|^2 + \|b\|^2)$, one gets,

$$g(\pi^\star) \leqslant \frac{2}{n} \sum_{i=1}^{n} \left\| (I_d - \hat{Q}^\top Q^\star) x_i \right\|^2 + \frac{2\sigma^2}{n} \sum_{i=1}^{n} \left\| \hat{Q}^\top z_i \right\|^2$$

$$= \frac{2}{n} \sum_{i=1}^{n} \left\| (\hat{Q} - Q^\star) x_i \right\|^2 + \frac{2\sigma^2}{n} \sum_{i=1}^{n} \|z_i\|^2,$$

where we used the fact that the matrices $\hat{Q}, Q^\star$ are orthogonal. Using concentration of Chi squared random variables, we have

$$\mathbb{P}\left( \sum_{i=1}^{n} \|z_i\|^2 \geqslant nd + 2\sqrt{ndt} + 2t \right) \leqslant e^{-t},$$

leading to $\mathbb{P}\left( \sum_{i=1}^{n} \|z_i\|^2 \geqslant 5nd \right) \leqslant e^{-nd}$ by plugging in $t = nd$. We are now left with $\sum_{i=1}^{n} \left\| (\hat{Q} - Q^\star) x_i \right\|^2$. We have that for any $Q$, $\mathbb{E}\left[ \sum_{i=1}^{n} \|(Q - Q^\star) x_i\|^2 \right] = n\|Q - Q^\star\|_F^2$; however, $\hat{Q}$ depends on the $x_i$ so we need a uniform upper bound. Using Hanson-Wright inequality, for any $Q \in \mathbb{R}^{d \times d}$,

$$\mathbb{P}\left( \sum_{i=1}^{n} \|(Q - Q^\star) x_i\|^2 \geqslant n\|Q - Q^\star\|_F^2 + c \max\left( \sqrt{nt\|Q - Q^\star\|_F^2 \|Q - Q^\star\|_{\mathrm{op}}^2}, t\|Q - Q^\star\|_{\mathrm{op}}^2 \right) \right) \leqslant 2e^{-t},$$

which reads as, for $Q$ orthogonal (leading to $\|Q - Q^\star\|_{\mathrm{op}} \leqslant 2$):

$$\mathbb{P}\left( \sum_{i=1}^{n} \|(Q - Q^\star) x_i\|^2 \geqslant n\|Q - Q^\star\|_F^2 + c' \max\left( \sqrt{nt\|Q - Q^\star\|_F^2}, t \right) \right) \leqslant 2e^{-t}.$$

For $\varepsilon \in (0, 1/2)$, let $\mathcal{N}_\varepsilon$ be an $\varepsilon-$net of $\mathcal{O}(d)$ of minimal cardinality; by Lemma 4 we have $\log(|\mathcal{N}_\varepsilon|) \leqslant Cd^2 \ln(d/\varepsilon)$. Using a union bound:

$$\mathbb{P}\left( \sup_{Q \in \mathcal{N}_\varepsilon} \left\{ \left| \sum_{i=1}^{n} \|(Q - Q^\star) x_i\|^2 - n\|Q - Q^\star\|_F^2 \right| \right\} \geqslant c' \max\left( \sqrt{nt\|Q - Q^\star\|_F^2}, t \right) \right) \leqslant 2e^{-t + Cd^2 \ln(d/\varepsilon)}.$$

Taking $t = \lambda + Cd^2 \ln(d/\varepsilon)$, we have with probabiliy $1 - 2e^{-\lambda}$ that:

$$\sup_{Q \in \mathcal{N}_\varepsilon} \left\{ \left| \sum_{i=1}^{n} \|(Q - Q^\star) x_i\|^2 - n\|Q - Q^\star\|_F^2 \right| \right\} \leqslant c' \max\left( \sqrt{n(\lambda + Cd^2 \ln(d/\varepsilon))\|Q - Q^\star\|_F^2}, \lambda + Cd^2 \ln(d/\varepsilon) \right)$$

Now, if $Q, Q' \in \mathcal{O}(d)$ satisfy $\|Q - Q'\|_F \leqslant \varepsilon$, we have using the orthogonality property of these matrices:

$$\sum_{i=1}^{n} \|(Q - Q^\star) x_i\|^2 - \sum_{i=1}^{n} \|(Q' - Q^\star) x_i\|^2 = 2\sum_{i=1}^{n} \langle (Q' - Q) x_i, Q^\star x_i \rangle$$

$$\leqslant 2\sum_{i=1}^{n} \|(Q' - Q) x_i\| \|Q^\star x_i\|$$

$$\leqslant 2\|Q' - Q\|_{\mathrm{op}} \sum_{i=1}^{n} \|x_i\|^2$$

$$\leqslant 2\varepsilon \sum_{i=1}^{n} \|x_i\|^2,$$

and with probability $1 - e^{-nd}$ we have $\sum_{i=1}^{n} \|x_i\|^2 \leqslant 5nd$. Then,

$$\|Q - Q^\star\|_F^2 - \|Q' - Q^\star\|_F^2 = \langle Q^\star, Q' - Q \rangle$$

$$\leqslant \|Q^\star\|_F \|Q' - Q\|_F$$

$$\leqslant \sqrt{d}\varepsilon.$$

Thus,

$$\sup_{Q\in\mathcal{O}(d)}\left\{\left|\sum_{i=1}^{n}\|(Q-Q^\star)x_i\|^2 - n\|Q-Q^\star\|_F^2\right|\right\} \leqslant \sup_{Q\in\mathcal{N}_\varepsilon}\left\{\left|\sum_{i=1}^{n}\|(Q-Q^\star)x_i\|^2 - n\|Q-Q^\star\|_F^2\right|\right\} + 10nd\varepsilon + n\sqrt{d}\varepsilon$$

$$\leqslant c'\max\left(\sqrt{n(\lambda + Cd^2\ln(d/\varepsilon))}\|Q-Q^\star\|_F^2, \lambda + Cd^2\ln(d/\varepsilon)\right)$$

$$+ 10nd\varepsilon + n\sqrt{d}\varepsilon,$$

with probability $1 - 2e^{-nd} - 2e^{-\lambda}$. Thus, applying this to $\hat{Q}$:

$$\sum_{i=1}^{n}\left\|(\hat{Q}-Q^\star)x_i\right\|^2 \leqslant n\delta d + c'\max\left(\sqrt{n\delta(\lambda + Cd^2\ln(11/\delta))}, \lambda + Cd^2\ln(1\varepsilon)\right) + 10nd\varepsilon + n\sqrt{d}\varepsilon,$$

and taking $\varepsilon = \delta/11$,

$$\sum_{i=1}^{n}\left\|(\hat{Q}-Q^\star)x_i\right\|^2 \leqslant 2n\delta d + c'\sqrt{n\delta(\lambda + Cd^2\ln(11/\delta))} + c'(\lambda + Cd^2\ln(11/\delta)),$$

leading to:

$$c^2(\hat{\pi}, \pi^\star) \leqslant 40\sigma^2 d + 16\delta d + c'\sqrt{\delta\frac{\lambda + Cd^2\ln(11/\delta)}{n}} + c'\frac{\lambda + Cd^2\ln(11/\delta)}{n},$$

hence the result, taking $\lambda = \sqrt{n} + d^2$.

$\square$

## B.2 Proof of Lemma 3

*Proof of Lemma 3.* Denote $g(Q) := \frac{1}{n}\sum_{i=1}^{n}\left\|x_{\hat{\pi}(i)} - Q^\top y_i\right\|^2$. The proof relies on noticing that for all $Q \in \mathcal{O}(d)$, by definition $g(\hat{Q}) \leqslant g(Q^\star)$ and using $\|a+b\|^2 \geqslant \frac{1}{2}\|a\|^2 - \|b\|^2$, one gets

$$g(\hat{Q}) \geqslant \frac{1}{2n}\sum_{i=1}^{n}\left\|(Q-\hat{Q})y_i\right\|^2 - g(Q),$$

and thus $\frac{1}{n}\sum_{i=1}^{n}\left\|(Q-\hat{Q})y_i\right\|^2 \leqslant 4g(Q^\star)$ by applying the previous inequality to $\pi = \pi^\star$. Using $\|a+b\|^2 \leqslant 2(\|a\|^2 + \|b\|^2)$, one gets:

$$g(Q^\star) = \frac{1}{n}\sum_{i=1}^{n}\left\|x_{\hat{\pi}(i)} - x_{\pi^\star(i)} - Q^{\star\top}z_i\right\|^2$$

$$\leqslant \frac{2}{n}\sum_{i=1}^{n}\left\|x_{\hat{\pi}(i)} - x_{\pi^\star(i)}\right\|^2 + \frac{2\sigma^2}{n}\sum_{i=1}^{n}\|z_i\|^2$$

$$= 2c^2(\hat{\pi}, \pi^\star) + \frac{2\sigma^2}{n}\sum_{i=1}^{n}\|z_i\|^2$$

$$\leqslant 2\delta d + \frac{2\sigma^2}{n}\sum_{i=1}^{n}\|z_i\|^2.$$

With probability $1 - e^{-nd}$, $\sum_{i=1}^{n}\|z_i\|^2 \leqslant 5nd$, and we are thus left with lower bounding $\frac{1}{n}\sum_{i=1}^{n}\left\|(Q-\hat{Q})y_i\right\|^2$. Using results form the previous proof, with probability $1 - 2e^{-nd} - 2e^{-\lambda}$ we have:

$$\frac{1}{1+\sigma^2}\sum_{i=1}^{n}\left\|(\hat{Q}-Q^\star)y_i\right\|^2 \geqslant n\left\|\hat{Q}-Q^\star\right\|_F^2 + n\delta d + c'\sqrt{n\delta(\lambda + Cd^2\ln(11/\delta))} + c'(\lambda + Cd^2\ln(11/\delta)).$$

Thus, with probability $1-$

$$\ell_{\text{ortho}}^2(\hat{Q}, Q^\star) \leqslant \delta d + c'\sqrt{n^{-1}\delta(\lambda + Cd^2\ln(11/\delta))} + c'n^{-1}(\lambda + Cd^2\ln(11/\delta)) + 8\delta d + 40\sigma^2 d,$$

leading to the desired result for $\lambda = d^2 + \sqrt{n}$. $\square$

## C  Proof of Theorem 1

*Proof of Theorem 1.* Define $\mathcal{L}(\pi, Q) := \frac{1}{n} \sum_{i=1}^{n} \left\| x_{\pi(i)} - Q^\top y_i \right\|^2$. Without loss of generality we can assume that $\pi^* = \mathrm{Id}$.

***Step 1: using ML estimators.*** By definition, the ML estimators $(\hat{\pi}, \hat{Q})$ defined in (2) minimize $\mathcal{L}$ and thus $\mathcal{L}(\hat{\pi}, \hat{Q}) \leqslant \mathcal{L}(\pi^\star = \mathrm{Id}, Q^\star)$, which can be expressed as:

$$\frac{1}{n} \sum_{i=1}^{n} \left\| x_{\hat{\pi}(i)} - \hat{Q}^\top Q^\star x_i - \sigma \hat{Q}^\top z_i \right\|^2 \leqslant \frac{\sigma^2}{n} \sum_{i=1}^{n} \|z_i\|^2.$$

Using $\|a\|^2 \leqslant 2(\|a - b\|^2 + \|b\|^2)$, we obtain:

$$\frac{1}{n} \sum_{i=1}^{n} \left\| x_{\hat{\pi}(i)} - \hat{Q}^\top Q^\star x_i \right\|^2 \leqslant \frac{4\sigma^2}{n} \sum_{i=1}^{n} \|z_i\|^2.$$

Then, standard chi-square concentration (see e.g. Laurent and Massart [2000]) entails that for all $t > 0$,

$$\mathbb{P}\left( \left| \sum_{i=1}^{n} \|z_i\|^2 - nd \right| \geqslant 2\sqrt{ndt} + 2t \right) \leqslant 2e^{-t},$$

so that with probability $1 - 2e^{-n}$, $\frac{4\sigma^2}{n} \sum_{i=1}^{n} \|z_i\|^2 \leqslant 4\sigma^2(d + 2\sqrt{d} + 2)$, and thus

$$\frac{1}{n} \sum_{i=1}^{n} \left\| x_{\hat{\pi}(i)} - \hat{Q}^\top Q^\star x_i \right\|^2 \leqslant 4\sigma^2(d + 2\sqrt{d} + 2) \leqslant 5\sigma^2 d, \tag{13}$$

for $d$ (or $n$) large enough.

***Step 2: existence of a set $\mathcal{K}$ with prescribed properties.*** We will now show that the above inequality (13) forces $\mathrm{ov}(\hat{\pi}, \pi^\star = \mathrm{Id})$ to be large. To do so, let us assume that $\mathrm{ov}(\hat{\pi}, \mathrm{Id}) < 1 - \delta$, for some $\delta > 0$ to be specified later: hence, there exist at least $n\delta$ indices $i \in [n]$ such that $\hat{\pi}(i) \neq i$. Let us define

$$\mathcal{I} := \left\{ i \in [n] : \left\| x_{\hat{\pi}(i)} - \hat{Q}^\top Q^\star x_i \right\|^2 \leqslant \frac{30}{\delta} \sigma^2 d \right\}.$$

It is clear that under the event $\mathcal{B}_n$ on which (13) holds, we have $(n - |\mathcal{I}|) \times \frac{30}{\delta} \sigma^2 d \leqslant 5n\sigma^2 d$, which gives

$$|\mathcal{I}| \geqslant n(1 - \delta/6).$$

Consequently, denoting

$$\mathcal{J} := \left\{ i \in [n] : \pi^\star(i) \neq \hat{\pi}(i), \left\| x_{\hat{\pi}(i)} - \hat{Q}^\top Q^\star x_i \right\|^2 \leqslant \frac{30}{\delta} \sigma^2 d \right\},$$

one has $|\mathcal{J}| \geqslant n\delta - (n - |\mathcal{I}|) \geqslant \frac{5}{6} \delta n$. We remark that for all $i \in \mathcal{J}$,

$$\left| \{ j \in \mathcal{J} : \{i, \hat{\pi}(i)\} \cap \{j, \hat{\pi}(j)\} \neq \varnothing \} \right| \leqslant 4.$$

Let us denote $Q := \hat{Q}^\top Q^\star$. Iteratively ruling out at most 3 elements for each $i \in \mathcal{J}$, the above shows that on event $\mathcal{E}_n$ one can build a set $\mathcal{K} := \mathcal{K}(\hat{\pi}, Q) \subseteq [n]$ such that

$(i)$  $|\mathcal{K}| \geqslant n\delta/6$,

$(ii)$  for all $i \in \mathcal{K}$, $\hat{\pi}(i) \neq i$, and $(i, \hat{\pi}(i))_{i \in \mathcal{K}}$ are disjoint pairs,

$(iii)$  for all $i \in \mathcal{K}$, $\left\| x_{\hat{\pi}(i)} - Q x_i \right\|^2 \leqslant \frac{30}{\delta} \sigma^2 d$.

***Step 3: upper bounding the probability of existence of such a set*** $\mathcal{K}$. We will now bound the probability that such a set $\mathcal{K}$ exists. First, let us fix $i \in [n]$, $Q \in \mathcal{O}(d)$, $\pi \in \mathcal{S}_n$ such that $\pi(i) \neq i$. We have $x_{\pi(i)} - Qx_i \sim \mathcal{N}(0, 2I_d)$. Assume $60\sigma^2 < 1$ and $\delta \geqslant 60\sigma^2$ so that we have $\frac{60}{\delta}\sigma^2 d \leqslant d$. For these fixed $Q, \pi$, we have

$$\mathbb{P}\left(\left\|x_{\pi(i)} - Qx_i\right\|^2 \leqslant \frac{60}{\delta}\sigma^2 d\right) \leqslant \mathbb{P}\left(\|\mathcal{N}(0, I_d)\|^2 \leqslant d/2\right) = \mathbb{P}\left(d - \|\mathcal{N}(0, I_d)\|^2 \geqslant d/2\right) \leqslant e^{-d/16},$$

where we applied the one-sided chi-square concentration inequality[4] $\mathbb{P}\left(k - X \geqslant 2\sqrt{kx}\right) \leqslant \exp(-x)$ when $X \sim \chi^2(k)$. This gives that for any given $\mathcal{K} \subset [n]$ satisfying conditions $(i)$ and $(ii)$ above, using independence of the pairs $(x_i, x_{\hat{\pi}(i)})_{i \in \mathcal{K}}$ and recalling that $\delta \geqslant 60\sigma^2$, one has

$$\mathbb{P}\left(\forall i \in \mathcal{K}, \left\|x_{\pi(i)} - Qx_i\right\|^2 \leqslant \frac{60}{\delta}\sigma^2 d\right) \leqslant e^{-n\delta/6 \times d/16} = e^{-\delta nd/96}. \tag{14}$$

Denote by $\mathcal{A}_\delta$ the event

$$\mathcal{A}_\delta := \{\text{there exists } \pi \in \mathcal{S}_n, Q \in \mathcal{O}(d) \text{ and } \mathcal{K} = \mathcal{K}(\pi, Q) \subseteq [n] \text{ which satisfies } (i), (ii) \text{ and } (iii)\}. \tag{15}$$

As previously explained, we want to bound the probability of the event $\mathcal{A}_\delta$, for $\delta \geqslant 60\sigma^2$. For the union bound on $Q \in \mathcal{O}(d)$, we need to use an epsilon-net argument, which is as follows. Let $\varepsilon > 0$ to be specified later. By Lemma 4, there exists $\mathcal{N}_\varepsilon$ an $\varepsilon-$net of $\mathcal{O}(d)$ of cardinality at most $\left(\frac{c_1\sqrt{d}}{\varepsilon}\right)^{d^2}$, which is also an $\varepsilon-$net for the operator norm, see Remark 2. In particular, if we are under event $A_\delta$ and take $\pi, Q, \mathcal{K}$ verifying conditions in (15), there exists an element $Q_\varepsilon$ of $O_\varepsilon(d)$ such that $\|Q_\varepsilon - Q\|_{op} \leqslant \varepsilon$, which gives

$$\forall i \in \mathcal{K}, \left\|x_{\pi(i)} - Q_\varepsilon x_i\right\| \leqslant \left\|x_{\pi(i)} - Qx_i\right\| + \|(Q - Q_\varepsilon)x_i\| \leqslant \sqrt{\frac{30}{\delta}\sigma^2 d} + \|Q_\varepsilon - Q\|_{op}\|x_i\|,$$

and applying chi-square concentration again gives that, under an event $\mathcal{C}_n$ with probability $\geqslant 1 - e^{-d + \log n} \geqslant 1 - e^{-d/2}$ since $(d \geqslant 2\log n)$ for all $i \in [n], \|x_i\| \leqslant \sqrt{2d}$ and the above yields

$$\forall i \in \mathcal{K}, \left\|x_{\pi(i)} - Q_\varepsilon x_i\right\| \leqslant \sqrt{\frac{30}{\delta}\sigma^2 d} + \varepsilon\sqrt{2d} \leqslant \sqrt{\frac{60}{\delta}\sigma^2 d},$$

choosing $\varepsilon = c_2\sqrt{\sigma^2/\delta}$ for some appropriate $c_2$. Hence, taking a union bound over $\pi \in \mathcal{S}_n, Q_\varepsilon \in O_\varepsilon(d)$ and subsets $\mathcal{K} \subseteq [n]$, and recalling (14), we can bound $\mathbb{P}\left(\mathcal{A}_\delta \mid \mathcal{C}_n, \mathcal{B}_n\right)$ by

$$\mathbb{P}\left(\mathcal{A}_\delta \mid \mathcal{B}_n, \mathcal{C}_n\right) \leqslant \frac{1}{\mathbb{P}\left(\mathcal{B}_n, \mathcal{C}_n\right)} \times n! \times \left(\frac{c_1\sqrt{d}}{\varepsilon}\right)^{d^2} \times 2^n \times e^{-\delta nd/96}$$

$$\leqslant (1 + o(1))\exp(n\log n + c_3 d^2\log d + c_4 d^2\sqrt{\delta/(\sigma^2)} + n\log 2 - c_5\delta nd)$$

$$\leqslant (1 + o(1))\exp(-c_6\delta nd)$$

where we recall that $\delta \geqslant 60\sigma^2$, and the last inequality holds if $c_4 d^2\sqrt{\delta/(\sigma^2)} \leqslant c_7 d^2 \leqslant c_8\delta nd$, and if $c_3 d^2\log d \leqslant c_8\delta nd$ for which $\delta \geqslant c_9 d\log d/n$ suffices, and if $n\log n \leqslant c_8\delta nd$, for which $\delta \geqslant c_10\log n/d$ suffices.

***Step 4: conclusion.*** Now, wrapping things up, we obtain that for $\delta \geqslant \max(60\sigma^2, c_9 d/n, c_10\log n/d)$, we have for $n$ large enough

$$\mathbb{P}\left(\text{ov}(\hat{\pi}, \pi^\star) < 1 - \delta\right) \leqslant \mathbb{P}\left(\mathcal{B}_n, \mathcal{C}_n\right)\mathbb{P}\left(\text{ov}(\hat{\pi}, \pi^\star) < 1 - \delta \mid \mathcal{B}_n, \mathcal{C}_n\right) + \mathbb{P}\left(\bar{\mathcal{B}}_n \cup \bar{\mathcal{C}}_n\right)$$

$$\leqslant \mathbb{P}\left(\mathcal{A}_\delta \mid \mathcal{B}_n, \mathcal{C}_n\right) + \mathbb{P}\left(\bar{\mathcal{B}}_n\right) + \mathbb{P}\left(\bar{\mathcal{C}}_n\right)$$

$$\leqslant (1 + o(1))e^{-c_6\delta nd} + 2e^{-n} + e^{-d/2} = o(1).$$

This gives the desired result

$$\text{ov}(\pi^\star, \hat{\pi}) \geqslant 1 - \max\left(60\sigma^2, c_1\frac{d}{n}, c_2\frac{\log n}{d\log d}\right),$$

which remains true when $60\sigma^2 \geqslant 1$.

The desired inequality for $\ell^2(\hat{Q}, Q^\star)$ follows from Lemma 3 (for $\delta = \Theta(d/n)$) and Remark 1. $\qquad\square$

---

[4] again, see e.g. Laurent and Massart [2000].

# D  Proof of Theorem 2

We here prove that a minimizer $\hat{Q}$ of the conic alignment loss satisfies the following guarantees.

**Theorem 3** (Conic alignment minimizer). *Let $\delta_0 \in (0,1)$. Let $q = p^3$. Let $v_1, \ldots, v_q$ be i.i.d. uniform directions in $\mathcal{S}^{d-1}$, and assume that $u_1, \ldots, u_p$ are independently and uniformly distributed over $\{v_1, \ldots, v_q\}$. Let $\mathcal{N}$ be an $\varepsilon-$net of $\mathcal{O}(d)$ for the Frobenius norm of minimal cardinality. Then, there exist constants $C_1, C_2, C_3, C_4, C_5 > 0$ such that, if $\log(n) \geqslant C_1 d \log(1/\delta_0)$, $\varepsilon = C_2 \sigma d^{-1/2}$, $\delta = \delta_0$, $\kappa = \sqrt{\frac{2}{d}}$, $p \geqslant \mathrm{polylog}(1/\sigma, d)$ and $\sigma \leqslant \frac{C_3 \delta_0^2}{\log(1/\delta_0)}$, then, with probability $1 - 6e^{-C_4 d^2}$,*

$$\frac{1}{d}\left\|\hat{Q} - Q^\star\right\|_F^2 \leqslant \delta_0\,.$$

Then, combinign this result with Lemma 2, setting $\hat{\pi}$ as in Equation (10), we obtain Theorem 2.

## D.1  Proof of Theorem 3

*Proof of Theorem 3.* Recall that $\delta, \kappa, \varepsilon > 0$ are for now any (small) positive number but can be specified later. $\delta_0$ is the target error. We begin by giving a few notations. For the proof, we need to introduce the following probability

$$\beta(\delta, \kappa) := \mathbb{P}\left(X \in \mathcal{C}(u, \delta), \|X\| \geqslant 1/\kappa\right)\,, \tag{16}$$

where $X \sim \mathcal{N}(0, I_d)$ and $u$ is any unit vector in $\mathbb{R}^d$. Note that $\beta(\delta, \kappa)$ is independent of the choice of $u$ by rotational invariance of Gaussian distribution. It is easy to check that $\mathbb{P}\left(x_i \in \mathcal{C}_\mathcal{X}(u, \delta)\right) = \mathbb{P}\left(y_j \in \mathcal{C}_\mathcal{X}(u, \delta)\right) = \beta(\delta, \kappa)$ for any $i, j$ and any unit vector $u$.

***Step 1: General strategy.*** Our goal is to prove that w.h.p. we have

$$F(\hat{Q}) < \inf\left\{F(Q), Q \in \mathcal{N}, \|Q - Q^\star\|_F^2 > \delta_0 d\right\}\,, \tag{17}$$

for some $\delta_0 > 0$ to be determined. This will entail that $\left\|\hat{Q} - Q^\star\right\|_F^2 \leqslant \delta_0 d$.

***Step 2: An upper bound on*** $\mathbb{E}\left[F(\hat{Q})\right]$. Since $\mathcal{N}$ is an $\varepsilon-$net of $\mathcal{O}(d)$, there exists $Q_\varepsilon^\star \in \mathcal{N}$ such that $\|Q_\varepsilon^\star - Q^\star\|_F \leqslant \varepsilon$. Note that by optimality of $\hat{Q}$, one has $F(\hat{Q}) \leqslant F(Q_\varepsilon^\star)$. We first upper bound the left hand side in (17) by upper bounding the expectation of $F(Q_\varepsilon^\star)$ using the following result.

**Lemma 5.** *Let $Q \in \mathcal{O}(d)$, $u \in \mathcal{S}^{d-1}$. We have:*

$$\mathbb{E}\left[F(Q)\right] \leqslant 2n\beta(\delta, \kappa)\left(c'd\left[\frac{2B\sigma}{\sqrt{6d\delta}} + \rho(Q^\star - Q)/\delta\right] + 2e^{-B^2/2} + e^{-d}\right)\,,$$

*for any $B^2 > 0$, where for some matrix $M$, $\rho(M)$ is defined as its spectral radius.*

Lemma 5 (proved in next subsection) gives, for any $B > 0$:

$$\begin{aligned}
F(\hat{Q}) \leqslant F(Q_\varepsilon^\star) &= \mathbb{E}\left[F(Q_\varepsilon^\star)\right] + F(Q_\varepsilon^\star) - \mathbb{E}\left[F(Q_\varepsilon^\star)\right] \\
&\leqslant 2n\beta(\delta, \kappa)\left(c'd\left[\frac{2B\sigma}{\sqrt{6d\delta}} + \rho(Q^\star - Q)/\delta\right] + 2e^{-B^2/2} + e^{-d}\right) + F(Q_\varepsilon^\star) - \mathbb{E}\left[F(Q_\varepsilon^\star)\right] \\
&\leqslant 2n\beta(\delta, \kappa)\left(c'd\left[\frac{2B\sigma}{\sqrt{6d\delta}} + \frac{\varepsilon}{\delta}\right] + 2e^{-B^2/2} + e^{-d}\right) + \sup_{Q \in \mathcal{N}}|F(Q) - \mathbb{E}\left[F(Q)\right]|\,, \tag{18}
\end{aligned}$$

where we used $\rho(Q^\star - Q_\varepsilon^\star) \leqslant \left\|\hat{Q} - Q_\varepsilon^\star\right\|_F \leqslant \varepsilon$ in the above.

***Step 3: A lower bound on*** $\mathbb{E}\left[F(Q)\right]$ ***for any*** $Q$. W lower bound the right hand side in (17) using the following Lemma.

**Lemma 6.** *Let $Q \in \mathcal{O}(d)$, $u \in \mathcal{S}^{d-1}$. We have, conditionally on the directions $u_1, \ldots, u_p$,*

$$\mathbb{E}\left[F(Q)\middle|u_1, \ldots, u_p\right] \geqslant 2C_1\beta(\delta, \kappa)\frac{\sum_{k=1}^p 1_{\left\{\|(Q^\star - Q)u_k\|_F^2 > 4\delta + 32\sigma\right\}}}{p}.$$

We recall that $\beta(\delta, \kappa)$ is defined in (16) here above. In the sequel, we denote by $\mathbb{P}_U$ (resp. $\mathbb{E}_U$) the probability (resp. expectation) over the directions $u_1, \ldots, u_p$. Lemma 6 (proved in next subsection) gives that

$$\inf\left\{F(Q), Q \in \mathcal{N}, \|Q - Q^\star\|_F^2 > \delta_0 d\right\} \geqslant \inf_{\substack{Q \in \mathcal{N} \\ \|Q - Q^\star\|_F^2 > \delta_0 d}} \mathbb{E}\left[F(Q)\right] - \sup_{Q \in \mathcal{N}} |F(Q) - \mathbb{E}\left[F(Q)\right]|$$

$$\geqslant \frac{2C_1 n\beta(\delta, \kappa)}{p} \inf_{\substack{Q \in \mathcal{N} \\ \|Q - Q^\star\|_F^2 > \delta_0 d}} \mathbb{E}_U\left[\sum_{k=1}^p 1_{\left\{\|(Q - Q^\star)u_k\|^2 > 4\delta + 32\sigma\right\}}\right] \tag{19}$$

$$- \sup_{Q \in \mathcal{N}} |F(Q) - \mathbb{E}\left[F(Q)\right]|. \tag{20}$$

Now, if we take $\delta_0 \geqslant 8\delta + 64\sigma$, we have

$$\inf_{\substack{Q \in \mathcal{N} \\ \|Q - Q^\star\|_F^2 > \delta_0 d}} \mathbb{E}_U\left[\sum_{k=1}^p 1_{\left\{\|(Q - Q^\star)u_k\|^2 > 4\delta + 32\sigma\right\}}\right] \tag{21}$$

$$\geqslant \inf_{\substack{Q \in \mathcal{N} \\ \|Q - Q^\star\|_F^2 > \delta_0 d}} \mathbb{E}_U\left[\sum_{k=1}^p 1_{\left\{\|(Q - Q^\star)u_k\|^2 > \delta_0/2\right\}}\right]$$

$$\geqslant \inf_{\substack{Q \in \mathcal{N} \\ \|Q - Q^\star\|_F^2 > \delta_0 d}} \mathbb{E}_U\left[\sum_{k=1}^p 1_{\left\{\|(Q - Q^\star)u_k\|^2 > \frac{\|Q - Q^\star\|_F^2}{2d}\right\}}\right]$$

$$= p \inf_{\substack{Q \in \mathcal{N} \\ \|Q - Q^\star\|_F^2 > \delta_0 d}} \mathbb{P}_U\left(\|(Q - Q^\star)u_1\|^2 > \frac{\|Q - Q^\star\|_F^2}{2d}\right).$$

Note that if $Z \sim \mathcal{N}(0, I_d/d)$, by rotational invariance of the Gaussian, one can always write $Z = Nu_1$ where $N = \|Z\|$ and $u_1 = \frac{Z}{\|Z\|}$ are independent and $u_1$ is uniform on the sphere. This yields $\frac{\|Q - Q^\star\|_F^2}{d} = \mathbb{E}\left[\|(Q - Q^\star)Z\|^2\right] = \mathbb{E}\left[N^2\right]\mathbb{E}_U[\|(Q - Q^\star)u_1\|^2] = 1 \times \mathbb{E}_U[\|(Q - Q^\star)u_1\|^2]$.

We can lower bound the right hand side of the above using a reverse Markov inequality, namely that $\mathbb{P}\left(X > \mathbb{E}\left[X\right]/2\right) \geqslant \mathbb{E}\left[X\right]/8$ for any $X$ such that $0 \leqslant X \leqslant 4$ a.s. We apply this to $X = \|(Q - Q^\star)u_1\|^2$ and get that for all $Q \in \mathcal{N}$ such that $\|Q - Q^\star\|_F^2 > \delta_0 d$,

$$\mathbb{P}_U\left(\|(Q - Q^\star)u_1\|^2 > \frac{\|Q - Q^\star\|_F^2}{2d}\right) \geqslant \frac{\|Q - Q^\star\|_F^2}{8d} \geqslant \frac{\delta_0}{8},$$

and via Equation (21), Equation (19) becomes

$$\inf\left\{F(Q), Q \in \mathcal{N}, \|Q - Q^\star\|_F^2 > \delta_0 d\right\} \geqslant \frac{C_1 n\beta(\delta, \kappa)\delta_0}{4} - \sup_{Q \in \mathcal{N}} |F(Q) - \mathbb{E}\left[F(Q)\right]|. \tag{22}$$

***Step 4: Uniform concentration of $F(Q)$ around its mean.*** The remaining step is to control the concentration of $F(Q)$ uniformly on $\mathcal{N}$. This is given by the following.

**Lemma 7** (Concentration of F)**.** *Let $Q \in \mathcal{O}(d)$ be fixed. Recall that $q = p^3$, that $v_1, \ldots, v_q$ are i.i.d. uniformly sampled on the sphere $\mathcal{S}^{d-1}$ and that $u_1, \ldots, u_p$ are i.i.d. uniformly sampled in $\{v_1, \ldots, v_q\}$. We have, for all $\lambda > 0$:*

$$\mathbb{P}\left(|F(Q) - \mathbb{E}\left[F(Q)\right]| \geqslant \frac{4\sqrt{2}\lambda\left(3\log(p) + \lambda + \frac{\log(q)^2 + \lambda^2}{9n\beta(\delta, \kappa)}\right)n\beta(\delta, \kappa)}{\sqrt{p}}\right) \leqslant 4e^{-\lambda} + 2e^{-\lambda^2}.$$

Hence, plugging $\lambda = 2\log(|\mathcal{N}|) > 1$ (assuming $|\mathcal{N}| \geqslant 2$) in Lemma 7, with probability at least $1 - \frac{6}{|\mathcal{N}|}$, we have

$$\sup_{Q \in \mathcal{N}} |F(Q) - \mathbb{E}\left[F(Q)\right]| \tag{23}$$

$$\leqslant 8\sqrt{2}\log(|\mathcal{N}|)\left(3\log(p) + 2\log(|\mathcal{N}|) + \frac{9\log(p)^2 + 4\log(|\mathcal{N}|)^2}{9n\beta(\delta,\kappa)}\right)n\beta(\delta,\kappa)p^{-1/2}$$

$$\leqslant c_4 d^2 \log(1/\varepsilon)\max(\log(p), d^2\log(1/\varepsilon))n\beta(\delta,\kappa)p^{-1/2} \tag{24}$$

$$+ c_5 d^2 \log(1/\varepsilon)\max(\log(p), d^2\log(1/\varepsilon))^2 p^{-1/2}\,.$$

where we used $\log(|\mathcal{N}|) \leqslant c_6 d^2 \log(1/\varepsilon)$ in the above.

***Step 5: Wrapping things up.*** Putting together the control on deviation in (23), the upper bound (18) and the lower bound (22), one gets that with probability $\geqslant 1 - 6/|\mathcal{N}|$, for any $B > 0$:

$$\inf\left\{F(Q), Q \in \mathcal{N}, \|Q - Q^\star\|_F^2 > \delta_0 d\right\} - F(\hat{Q}) \tag{25}$$

$$\geqslant \frac{C_1 n\beta(\delta,\kappa)\delta_0}{4} - 2n\beta(\delta,\kappa)\left(c'd\left[\frac{2B\sigma}{\sqrt{6d}\delta} + \frac{\varepsilon}{\delta}\right] + 2e^{-B^2/2} + e^{-d}\right)$$

$$- 2\sup_{Q \in \mathcal{N}} |F(Q) - \mathbb{E}\left[F(Q)\right]|$$

$$\geqslant n\beta(\delta,\kappa)\frac{C_1\delta_0}{4}$$

$$- n\beta(\delta,\kappa)\left(c'd\left[\frac{2B\sigma}{\sqrt{6d}\delta} + \frac{\varepsilon}{\delta}\right] + 2e^{-B^2/2} + e^{-d} - c_4 d^2\log(1/\varepsilon)\max(\log(p), d^2\log(1/\varepsilon))p^{-1/2}\right)$$

$$- c_5 d^2\log(1/\varepsilon)\max(\log(p), d^2\log(1/\varepsilon))^2 p^{-1/2}\,. \tag{26}$$

If this lower bound is positive, we can conclude that $\left\|\hat{Q} - Q^\star\right\|_F^2 \leqslant \delta_0 d$, as desired.

So far, the only constraints on our constants are

$$\text{(A1)} \quad \delta_0 \geqslant 8\delta + 64\sigma\,.$$

While the noise parameter $\sigma$ is fixed in this proof, we have the freedom to impose some constraints on $\varepsilon$ (the granularity of the $\varepsilon-$net), $\delta$ (the width of the cones), $\kappa$ (the truncature parameter), $p$ (the number of directions) and $B$ in order to make the expression in (25) positive (and even $\gg 1$). The remaining step is to show that this is possible ; this is what we shall do now.

Recall that we are in a regime where we need to keep$n$ in our min$f$ that $n$ tends to $+\infty$ and $d$ tends to $+\infty$ with $n$ but with $d \leqslant \log(n)$. We want to show that the positive term in (25) can dominate the others ; first, we would like to have an inequality of the form

$$\frac{C_1\delta_0}{4} - c'd\left[\frac{2B\sigma}{\sqrt{6d}\delta} + \frac{\varepsilon}{\delta}\right] + 2e^{-B^2/2} + e^{-d} - c_4 d^2\log(1/\varepsilon)\max(\log(p), d^2\log(1/\varepsilon))p^{-1/2} > c_6\delta_0\,,$$
$$\tag{27}$$

which is going to be satisfied if:

- (A2) $C_1 > 8c_6$,
- (A3) $\delta_0\delta \geqslant c_7\sqrt{d}\sigma B$,
- (A4) $\delta_0\delta \geqslant c_8 d\varepsilon$,
- (A5) $\delta_0 \geqslant c_9(e^{-B^2/2} + e^{-d})$,
- (A6) $\delta_0 \geqslant c_{10}d^2\log(1/\varepsilon)\max(\log(p), d^2\log(1/\varepsilon))p^{-1/2}$,

where $c_7, c_8, c_9, c_{10}$ are large enough constants. Now,

- (A2) is easily verified by choosing $c_6$.

- $B$ only appears in (A3) and (A5). (A5) is satisfied if we take $\delta_0 \geqslant 2c_9 e^{-d}$ and $B^2 = c_{11} \max(1, \log(1/\delta_0))$, transforming (A3) into $\frac{\delta_0 \delta}{\sqrt{\max(1, \log(1/\delta_0))}} \geqslant c_{12} \sqrt{d}\sigma$;

- $\varepsilon$ appears in (A4) and can be taken as $\varepsilon \leqslant \frac{\delta_0 \delta}{c_8 d}$ for this condition to be satisfied. Combined with (A2), we can simply take $\varepsilon \leqslant c_7 \sqrt{d}\sigma B/(c_8 d)$ for this condition to be redundant;

- $p$ only appears in (A6) and can thus be taken as large as desired to have this inequality satisfied (very large $p$ does not degrade any bound).

Consequently, the inequality in Equation (27) is satisfied for parameters that satisfy:

$$\text{(A7)} \quad \varepsilon = c_{16} d^{-1/2} \sigma\,, \qquad\qquad \text{(A8)} \quad p^{1/2} \geqslant c_{13} \sigma^{-1} d^{7/2} \log(d/\sigma)^3\,,$$

$$\text{(A9)} \quad \delta = \delta_0\,, \qquad\qquad \text{(A10)} \quad \frac{\delta_0^2}{\max(1, \log(1/\delta_0))} \geqslant c_{15} \sqrt{d}\sigma\,,$$

thereby transforming the condition that the RHS in Equation (25) is positive into:

$$n\beta(\delta, \kappa) \geqslant c_{16} d^2 \log(1/\varepsilon) \max(\log(p), d^2 \log(1/\varepsilon))^2 p^{-1/2} \left[\sqrt{d}\sigma \log(\sqrt{d}\sigma)\right]^{-1}\,.$$

The RHS of this inequality can be taken smaller than 1 by imposing that $p$ is large enough (recall that $p$ can be taken as large as desired). The final condition thus reads as $n\beta(\delta, \kappa) \geqslant 1$, which is itself satisfied if

$$n \geqslant e^{c'd \log(1/\delta)} (1 - e^{-d/16})^{-1}\,,$$

since $\beta(\delta, \kappa) = \mathbb{P}(x_1 \in \mathcal{C}_{\mathcal{X}}(u_0, \delta), \|x_1\| \geqslant 1/\kappa)$ and $\mathbb{P}(x_1 \in \mathcal{C}_{\mathcal{X}}(u_0, \delta)) \geqslant e^{-c'd \log(1/\delta)}$, while $\mathbb{P}(\|x_1\| \geqslant 1/\kappa) \geqslant 1 - e^{-d/16}$ for

$$\text{(A12)} \quad \kappa^2 = \frac{2}{d}\,.$$

We are now going to use the low-dimensionality assumption $d \ll \log(n)$, since $n \geqslant e^{c'd \log(1/\delta)}(1 - e^{-d/16})$ will be verified for

$$\text{(A13)} \quad \log(n) \geqslant c''d \log(1/\delta) = c''d \log(1/\delta_0)\,.$$

Thus, under (A8-A13), Equation (25) is positive, and therefore we have that $\frac{1}{d}\left\|\hat{Q} - Q^\star\right\|_F^2 \leqslant \delta_0$. $\quad\square$

## D.2 Misceallenous lemmas on the path to proving Theorem 3

We introduce the (numerical) constants $c, c' > 0$ that verify, for all $\delta' \in (0, 1/4)$ that for $x$ sampled uniformly on $\mathcal{S}^{d-1}$ and any $u \in \mathcal{S}^{d-1}$ we have[5]:

$$\exp\left(-c'd \log(1/\delta')\right) \leqslant \mathbb{P}\left(x \in \mathcal{C}(u, \delta')\right) \leqslant \exp\left(-cd \log(1/\delta')\right)\,.$$

The following lemmas are used to prove Lemma 6 and Lemma 5.

---

[5]In our model, the probability $\mathbb{P}(x \in \mathcal{C}(u, \delta'))$ can in fact be computed explicitly. For fixed $d$ and $n$, the above probability is given by $(1/2)\mathbb{P}\left(X_1^2 \geqslant (1-\delta)^2(X_1^2 + \ldots X_d^2)\right)$ where the $(X_i)$ are standard i.i.d. Gaussian variables. It is standard that $\frac{X_1^2}{\|X\|^2}$ is distributed according to the beta distribution $\beta(1/2, (d-1)/2)$, hence

$$\mathbb{P}\left(x \in \mathcal{C}(u, \delta')\right) = \frac{1}{2}\mathbb{P}\left(\beta(1/2, (d-1)/2) \geqslant (1-\delta)^2\right) = \frac{\Gamma(d/2)}{\Gamma(1/2)\Gamma(\frac{d-1}{2})} \int_{(1-\delta)^2}^1 x^{-1/2}(1-x)^{(d-3)/2} dx\,,$$

which is indeed of order $c\delta^d$ when $\delta$ is small.

*Proof of Lemma 5.* We recall that for any $i, j$, $\mathbb{P}\left(x_i \in \mathcal{C}_{\mathcal{X}}(u, \delta)\right) = \mathbb{P}\left(y_j \in \mathcal{C}_{\mathcal{X}}(u, \delta)\right) = \beta(\delta, \kappa)$ for any unit vector $u$. Taking the expectation and developing the indicators, we have

$$
\mathbb{E}\left[F(Q)\right]
$$
$$
= \frac{1}{p} \sum_{k=1}^{p} \sum_{i=1}^{n} \left(\mathbb{P}\left(x_{\pi^\star(i)} \in \mathcal{C}_{\mathcal{X}}(Qu_k, \delta)\right) + \mathbb{P}\left(y_i \in \mathcal{C}_{\mathcal{Y}}(u_k, \delta)\right) - 2\mathbb{P}\left(x_{\pi^\star(i)} \in \mathcal{C}_{\mathcal{X}}(Qu_k, \delta) | y_i \in \mathcal{C}_{\mathcal{Y}}(u_k, \delta)\right) \mathbb{P}\left(y_i \in \mathcal{C}_{\mathcal{Y}}(u_k, \delta)\right)\right.
$$
$$
= \frac{2\beta(\delta, \kappa)}{p} \sum_{k=1}^{p} \sum_{i=1}^{n} \left(1 - \mathbb{P}\left(x_{\pi^\star(i)} \in \mathcal{C}(Qu_k, \delta), \left\|x_{\pi^\star(i)}\right\| \geqslant 1/\kappa \Big| y_i \in \mathcal{C}(u_k, \delta), \|y_i\| \geqslant \sqrt{1+\sigma^2}/\kappa\right)\right)
$$
$$
= 2n\beta(\delta, \kappa) \left(1 - \mathbb{P}\left(X \in \mathcal{C}(Qu, \delta), \|X\| \geqslant 1/\kappa \Big| Y \in \mathcal{C}(u, \delta), \|Y\| \geqslant \sqrt{1+\sigma^2}/\kappa\right)\right),
$$

where $X \sim \mathcal{N}(0, I_d)$, $Y = Q^\star X + \sigma Z$ with $Z \sim \mathcal{N}(0, I_d)$ independent from $X$, and for any unit vector $u$. We thus need to bound the last term, which is done by noticing that the two events in the remaining probability become highly positively correlated when $Q$ is close to $Q^\star$. First, we separate the norm component from the direction component in the event $\{X \in \mathcal{C}(Qu, \delta), \|X\| \geqslant 1/\kappa\}$:

$$
\mathbb{P}\left(X \in \mathcal{C}(Qu, \delta), \|X\| \geqslant 1/\kappa \Big| Y \in \mathcal{C}(u, \delta), \|Y\| \geqslant \sqrt{1+\sigma^2}/\kappa\right)
$$
$$
= \mathbb{P}\left(X \in \mathcal{C}(Qu, \delta) \Big| Y \in \mathcal{C}(u, \delta), \|Y\| \geqslant \sqrt{1+\sigma^2}/\kappa\right) \mathbb{P}\left(\|X\| \geqslant 1/\kappa \big| \|Y\| \geqslant \sqrt{1+\sigma^2}/\kappa\right)
$$
$$
\geqslant \mathbb{P}\left(X \in \mathcal{C}(Qu, \delta) \Big| Y \in \mathcal{C}(u, \delta), \|Y\| \geqslant \sqrt{1+\sigma^2}/\kappa\right) \mathbb{P}\left(\|X\| \geqslant 1/\kappa\right).
$$

Now, we have $y_i \in \mathcal{C}_{\mathcal{Y}}(u, \delta) \implies x_{\pi^\star(i)} \in \mathcal{C}_{\mathcal{X}}(Q^\top u, \delta + \delta_i(Q, u))$ using Lemma 12, where $\delta_i(Q) = 2\sigma \frac{|\langle z_i, (Q^\star)^\top u\rangle|}{\|x_{\pi^\star(i)}\|} + \rho(Q^\star - Q)$, so that:

$$
\mathbb{P}\left(x_{\pi^\star(i)} \in \mathcal{C}(Q, \delta) \Big| y_i \in \mathcal{C}(u, \delta), \|y_i\| \geqslant \sqrt{1+\sigma^2}/\kappa\right)
$$
$$
= \mathbb{P}\left(x_{\pi^\star(i)} \in \mathcal{C}(Q, \delta) \Big| y_i \in \mathcal{C}(u, \delta), x_{\pi^\star(i)} \in \mathcal{C}(Q, \delta + \delta_i(Q, u)), \|y_i\| \geqslant \sqrt{1+\sigma^2}/\kappa\right)
$$
$$
\geqslant \mathbb{E}\left[\exp\left(-c'd \log\left(1 + 2\sigma \frac{|\langle z_i, (Q^\star)^\top u\rangle|}{\delta \|x_{\pi^\star(i)}\|} + \rho(Q^\star - Q)/\delta\right)\right) \Big| \|y_i\| \geqslant \sqrt{1+\sigma^2}/\kappa\right]
$$
$$
\geqslant \mathbb{E}\left[\exp\left(-c'd \log\left(1 + \frac{2\sigma B'}{\delta \|x_{\pi^\star(i)}\|} + \rho(Q^\star - Q)/\delta\right)\right) \Big| \|y_i\| \geqslant \sqrt{1+\sigma^2}/\kappa, |\langle z_i, (Q^\star)^\top u\rangle| \leqslant B'\right]
$$
$$
\times \mathbb{P}\left(|\langle z_i, (Q^\star)^\top u\rangle| \leqslant B'\right)
$$
$$
\geqslant \exp\left(-c'd \log\left(1 + \frac{2\kappa\sigma B}{\delta} + \rho(Q^\star - Q)/\delta\right)\right) (1 - \mathbb{P}\left(|\langle z_i, (Q^\star)^\top u\rangle| > B\right))(1 - \mathbb{P}\left(\|x_{\pi^\star(i)}\| \geqslant 1/\kappa\right)).
$$

First, $\mathbb{P}\left(\left\|x_{\pi^\star(i)}\right\|^2 > d + 2\sqrt{dt} + t\right) \leqslant e^{-t}$ for any $t > 0$, so that if $1/\kappa^2 \geqslant 3d$, $\mathbb{P}\left(\left\|x_{\pi^\star(i)}\right\| > 1/\kappa\right) \leqslant e^{-\frac{1}{3\kappa^2} + d}$.

Then, $|\langle z_i, (Q^\star)^\top u\rangle| \sim |\mathcal{N}(0, 1)|$ since $u$ us unitary, and thus $\mathbb{P}\left(|\langle z_i, (Q^\star)^\top u\rangle| > B\right) \leqslant 2e^{-B^2/2} \leqslant 2e^{-2}$ for $B = 2$, leading to:

$$
\mathbb{P}\left(x_{\pi^\star(i)} \in \mathcal{C}(Q, \delta) \Big| y_i \in \mathcal{C}(u, \delta), \|y_i\| \geqslant \sqrt{1+\sigma^2}/\kappa\right)
$$
$$
\geqslant \exp\left(-c'd \log\left(1 + \frac{2B\kappa\sigma}{\delta} + \rho(Q^\star - Q)/\delta\right)\right) (1 - 2e^{-B^2/2})(1 - e^{-\frac{1}{3\kappa^2} + d})
$$
$$
\geqslant \exp\left(-c'd \log\left(1 + \frac{2N\sigma}{\sqrt{6d}\delta} + \rho(Q^\star - Q)/\delta\right)\right) (1 - 2e^{-B^2/2} - e^{-d}),
$$

for $\kappa^2 = 1/(6d)$. Using $\log(1+x) \leqslant x$ and $e^{-x} \geqslant 1-x$ for $x \geqslant 0$,

$$\mathbb{P}\left(x_{\pi^\star(i)} \in \mathcal{C}(Q,\delta) \Big| y_i \in \mathcal{C}(u,\delta), \|y_i\| \geqslant \sqrt{1+\sigma^2}/\kappa\right)$$

$$\geqslant \exp\left(-c'd\big(\frac{2B\sigma}{\sqrt{6d}\delta} + \rho(Q^\star - Q)/\delta\big)\right)(1 - 2e^{-B^2/2} - e^{-d})$$

$$\geqslant \left(1 - c'd\big(\frac{2B\sigma}{\sqrt{6d}\delta} + \rho(Q^\star - Q)/\delta\big)\right)(1 - 2e^{-B^2/2} - e^{-d})$$

$$\geqslant \left(1 - c'd\left[\frac{2B\sigma}{\sqrt{6d}\delta} + \rho(Q^\star - Q)/\delta\right] - 2e^{-B^2/2} - e^{-d}\right).$$

leading to

$$1 - \mathbb{P}\left(x_{\pi^\star(i)} \in \mathcal{C}(Q,\delta) \Big| y_i \in \mathcal{C}(u,\delta), \|y_i\| \geqslant \sqrt{1+\sigma^2}/\kappa\right)$$

$$\leqslant c'd\left[\frac{2B\sigma}{\sqrt{6d}\delta} + \rho(Q^\star - Q)/\delta\right] + 2e^{-B^2/2} + e^{-d},$$

and thus to the desired upper bound on $\mathbb{E}\left[F(Q)\right]$. $\qquad\square$

*Proof of Lemma 7.* We first begin by bounding all the terms that appear in the sum of $F(Q)$. Define

$$A(u, Q) = |\mathcal{C}_\mathcal{X}(Qu, \delta)| - |\mathcal{C}_\mathcal{Y}(u, \delta)|,$$

so that $F(Q) = \frac{1}{p}\sum_{k=1}^p A(u_k, Q)^2$. Using Bernstein inequality [Vershynin, 2018, Theorem 2.8.4], and writing $\beta(\delta, \kappa) = \mathbb{P}\left(x_{\pi^\star(i)} \in \mathcal{C}_\mathcal{X}(Qu, \delta)\right)$ (so that $\mathbb{E}\left[A(u, Q)\right] \leqslant n\beta(\delta, \kappa)$), we have:

$$\mathbb{P}\left(|A(u, Q)| \geqslant t\right) \leqslant 2\exp\left(-\frac{t^2/2}{n\beta(\delta, \kappa) + t/3}\right),$$

so that

$$\mathbb{P}\left(A(u, Q)^2 \geqslant n\beta(\delta, \kappa)t\right) \leqslant 2\exp\left(-\frac{n\beta(\delta, \kappa)t/2}{n\beta(\delta, \kappa) + \sqrt{n\beta(\delta, \kappa)t}/3}\right)$$

$$\leqslant 2\exp\left(-\frac{t}{4}\right) + 2\exp\left(-\frac{3\sqrt{n\beta(\delta, \kappa)t}}{2}\right).$$

Now, we have:

$$\mathbb{P}\left(\exists \ell \in [q], \quad A(v_\ell, Q)^2 \geqslant n\beta(\delta, \kappa)t\right) \leqslant 2\exp\left(-\frac{t}{4} + \log(q)\right) + 2\exp\left(-\frac{3\sqrt{n\beta(\delta, \kappa)t}}{2} + \log(q)\right)$$

$$= 4e^{-\lambda},$$

for $t = 4(3\log(p) + \lambda) + \frac{4(\log(q)^2 + \lambda^2)}{9n\beta(\delta, \kappa)}$. We now use MacDiarmid's inequality [Vershynin, 2018, Theorem 2.9.1], by seeing $F(Q)$ as $F(Q) = f(u_1, \ldots, u_p)$, conditionally on the event $\forall \ell \in [q], A(v_\ell, Q)^2 \leqslant \left(4(3\log(p) + \lambda) + \frac{4(\log(q)^2 + \lambda^2)}{9n\beta(\delta, \kappa)}\right)n\beta(\delta, \kappa) = B$, to obtain

$$\mathbb{P}\left(\left|F(Q) - \mathbb{E}\left[F(Q)|V\right]\right| \geqslant t \Big| \forall \ell \in [q], A(v_\ell, Q)^2 \leqslant B\right) \leqslant 2\exp\left(\frac{pt^2}{2B^2}\right),$$

where $V = \{v_1, \ldots, v_q\}$, since the bounded difference inequality is then verified for constant $4B$. Thus,

$$\mathbb{P}\left(\left|F(Q) - \mathbb{E}\left[F(Q)|V\right]\right| \geqslant \frac{\sqrt{2}\left(4(3\log(p) + \lambda) + \frac{4(\log(q)^2 + \lambda^2)}{9n\beta(\delta, \kappa)}\right)n\beta(\delta, \kappa)}{\sqrt{p}}\right) \leqslant 4e^{-\lambda} + 2e^{-\lambda^2}.$$

The problem here lies in the fact that $\mathbb{E}[F(Q)|V] = \mathbb{E}[F(Q)]$ may not always hold! Hopefully this is in fact the case:

$$\mathbb{E}[F(Q)|V] = \frac{1}{pq}\sum_{k=1}^{p}\sum_{\ell=1}^{q}\mathbb{E}\left[(|\mathcal{C}_{\mathcal{X}}(Qv_\ell,\delta)| - |\mathcal{C}_{\mathcal{Y}}(v_\ell,\delta)|)^2|u_k = v_\ell\right]$$
$$= \mathbb{E}\left[(|\mathcal{C}_{\mathcal{X}}(Qv,\delta)| - |\mathcal{C}_{\mathcal{Y}}(v,\delta)|)^2\right] \qquad \text{for any fixed } v \in \mathcal{S}^{d-1}$$
$$= \mathbb{E}[F(Q)],$$

concluding the proof. $\qquad\square$

*Proof of Lemma 6.* Let $\varepsilon > 0$ to be determined later and $k \in [p]$ such that $\|(Q-Q^\star)u_k\| > \varepsilon$. We are going to show that $\mathbb{P}\left(x_{\pi^\star(i)} \in \mathcal{C}(Qu_k,\delta)\middle| y_i \in \mathcal{C}(u_k,\delta), \|y_i\| \geqslant \sqrt{1+\sigma^2}/\kappa, \|(Q-Q^\star)u_k\| > \varepsilon\right)$ is small.

Using Lemma 11, $y_i \in \mathcal{C}(u_k,\delta)$ implies that $x_{\pi^\star(i)} \in \mathcal{C}(Q^\star u_k, \delta + 2\frac{\sigma\|z_i\|}{\|x_{\pi^\star(i)}\|})$. Then, $\mathcal{C}(Qu_k,\delta) \cap \mathcal{C}(Q^\star u_k, \delta + 2\frac{\sigma\|z_i\|}{\|x_{\pi^\star(i)}\|}) = \varnothing$ provided that $\|Qu_k - Q^\star u_k\|^2 > 4(\delta + \frac{\sigma\|z_i\|}{\|x_{\pi^\star(i)}\|})$ using Lemma 8. Thus, if $\|Qu_k - Q^\star u_k\|^2 > \varepsilon$,

$$\mathbb{P}\left(x_{\pi^\star(i)} \in \mathcal{C}(Qu_k,\delta)\middle| y_i \in \mathcal{C}(u_k,\delta), \|y_i\| \geqslant \sqrt{1+\sigma^2}/\kappa, \|(Q-Q^\star)u_k\| > \varepsilon\right)$$
$$\leqslant \mathbb{P}\left(4(\delta + \frac{\sigma\|z_i\|}{\|x_{\pi^\star(i)}\|}) > \varepsilon\right)$$
$$\leqslant \mathbb{P}\left(\frac{4\sigma\|z_i\|}{\|x_{\pi^\star(i)}\|} > \varepsilon - 4\delta\right).$$

We have $\mathbb{P}\left(\|z_i\|^2 \geqslant 4d\right) \leqslant e^{-d}$, and $\mathbb{P}\left(\|x_{\pi^\star(i)}\|^2 \leqslant \frac{d}{2}\right) \leqslant e^{-d/16}$, so that if $\varepsilon \geqslant 4\delta + 32\sigma$, we have $\mathbb{P}\left(\frac{4\sigma\|z_i\|}{\|x_{\pi^\star(i)}\|} > \varepsilon - 4\delta\right) \leqslant \mathbb{P}\left(\|z_i\|^2 \geqslant 4d\right) + \mathbb{P}\left(\|x_{\pi^\star(i)}\|^2 \leqslant \frac{d}{2}\right) \leqslant e^{-d} + e^{-16d}$, leading to

$$\mathbb{P}\left(x_{\pi^\star(i)} \in \mathcal{C}(Qu_k,\delta)\middle| y_i \in \mathcal{C}(u_k,\delta), \|y_i\| \geqslant \sqrt{1+\sigma^2}/\kappa, \|(Q-Q^\star)u_k\| > \varepsilon\right) \leqslant e^{-d} + e^{-d/16} \leqslant 1 - C_1,$$

where $C_1 = 1/e + 1/e^{1/16} > 0$ is a numerical constant. This thus gives:

$$\mathbb{E}[F(Q)|U] \geqslant 2C_1\beta(\delta,\kappa)\frac{\sum_{k=1}^{p}\mathbb{1}_{\left\{\|(Q^\star-Q)u_k\|_F^2 > \varepsilon\right\}}}{p}.$$
$$\square$$

**Lemma 8** (Cone separation). *For $u,v \in \mathcal{S}^{d-1}$, $\mathcal{C}(u,\delta) \cap \mathcal{C}(v,\delta) \neq \varnothing$ implies that $\|u-v\|^2 \leqslant 8\delta$.*

*Proof.* Assume $\mathcal{C}(u,\delta) \cap \mathcal{C}(v,\delta) \neq \varnothing$. Take $w \in \mathcal{C}(u,\delta) \cap \mathcal{C}(v,\delta)$: we can always assume that $\|w\| = 1$ by rescaling. Then, by triangle inequality, we have $\|u-v\| \leqslant \|u-w\| + \|v-w\|$. Since $w \in \mathcal{C}(u,\delta)$, $\|u-w\|^2 = 2 - 2\langle v,w\rangle \leqslant 2 - 2(1-\delta) = 2\delta$, and the same is true for $\|v-w\|$. This gives $\|u-v\| \leqslant 2\sqrt{2\delta}$. $\qquad\square$

**Lemma 9** (Probability that two cones are disjoint). *Let $Q,Q' \in \mathcal{O}(d)$, $\delta \leqslant \frac{1}{12d}\|Q'-Q\|_F^2$ and let $u$ be a random variable uniformly distributed over $\mathcal{S}^{d-1}$. Then,*
$$\mathbb{P}\left(\mathcal{C}(Q'u,\delta) \cap \mathcal{C}(Qu,\delta) = \varnothing\right) \geqslant \delta.$$

*Proof.* Using the previous Lemma, $\mathbb{P}\left(\mathcal{C}(Q'u,\delta) \cap \mathcal{C}(Qu,\delta) \neq \varnothing\right) \leqslant \mathbb{P}\left(\|Qu - Q'u\|^2 \leqslant 8\delta\right)$. Let $Z$ be the random variable $Z = \|Qu - Q'u\|^2$. We have that $\mathbb{E}[Z] = \|Q-Q'\|_F^2/d \geqslant 12\delta$ and $Z \leqslant 4$ almost surely. Thus, using a "reverse Markov" inequality,

$$12\delta \leqslant \mathbb{E}[Z] = \mathbb{E}[Z\mathbb{1}_{Z\leqslant 8\delta}]\mathbb{P}(Z\leqslant 8\delta) + \mathbb{E}[Z\mathbb{1}_{Z>8\delta}]\mathbb{P}(Z>8\delta)$$
$$\leqslant 8\delta\mathbb{P}(Z\leqslant 8\delta) + 4\mathbb{P}(Z>8\delta)$$
$$\leqslant 8\delta + 4(1 - \mathbb{P}(X\leqslant 8\delta)),$$

that is $\mathbb{P}(X \leqslant 8\delta) \leqslant 1 - \delta$, which concludes the proof. $\qquad\square$

The following Lemma is easy and does require any proof.

**Lemma 10.** *For any* $u$, $\delta \in (0,1)$, *we have* $\mathbb{E}\left[|\mathcal{C}_{\mathcal{X}}(u,\delta)|\right] = \mathbb{E}\left[|\mathcal{C}_{\mathcal{Y}}(u,\delta)|\right] = n\mathbb{P}\left(x_1 \in \mathcal{C}(u,\delta), \|x_1\| \geqslant 1/\kappa\right) = n\beta(\delta,\kappa)$ *so that* $|\mathcal{C}_{\mathcal{X}}(Qu,\delta)| - |\mathcal{C}_{\mathcal{Y}}(u,\delta)|$ *in the sum that defines* $F$ *are all centered.*

$|\mathcal{C}_{\mathcal{X}}(u,\delta)|$ *and* $|\mathcal{C}_{\mathcal{Y}}(u,\delta)|$ *are (correlated) binomial random variables of parameters* $(n, \beta(\delta,\kappa))$.

**Lemma 11.** *For any* $u \in \mathcal{S}^{d-1}$, $i \in [n]$, *we have* $y_i \in \mathcal{C}_{\mathcal{Y}}(u,\delta) \implies x_{\pi^\star(i)} \in \mathcal{C}_{\mathcal{X}}((Q^\star)^\top u, \delta + \delta_i)$, *where* $\delta_i = 2\sigma \frac{\|z_i\|}{\|x_{\pi^\star(i)}\|}$ *and* $x_{\pi^\star(i)} \in \mathcal{C}_{\mathcal{X}}((Q^\star)^\top u, \delta + \delta_i') \implies y_i \in \mathcal{C}_{\mathcal{Y}}(u,\delta)$, *where* $\delta_i' = 2\sigma \frac{\|z_i\|}{\|y_i\|}$.

*Proof.* Let us prove the first assertion and assume that $y_i \in \mathcal{C}_{\mathcal{Y}}(u,\delta)$. We have $y_i = Q^\star x_{\pi^\star(i)} + \sigma z_i \in \mathcal{C}_{\mathcal{Y}}(u,\delta)$, which writes as:

$$\langle Q^\star x_{\pi^\star(i)} + \sigma z_i, u \rangle \geqslant (1-\delta)\|Q^\star x_{\pi^\star(i)} + \sigma z_i\|.$$

Thus,

$$
\begin{aligned}
\langle x_{\pi^\star(i)}, (Q^\star)^\top u \rangle &\geqslant (1-\delta)\|Q^\star x_{\pi^\star(i)} + \sigma z_i\| - \sigma\langle z_i, (Q^\star)^\top u \rangle \\
&\geqslant (1-\delta)\|Q^\star x_{\pi^\star(i)} + \sigma z_i\| - \sigma\|z_i\| \\
&\geqslant (1-\delta)(\|Q^\star x_{\pi^\star(i)}\| - \sigma\|z_i\|) - \sigma\|z_i\| \\
&\geqslant (1-\delta)\|Q^\star x_{\pi^\star(i)}\| - 2\sigma\|z_i\| \\
&\geqslant (1-\delta-\delta_i)\|Q^\star x_{\pi^\star(i)}\|,
\end{aligned}
$$

which is the desired result. The second assertion is proved exactly in the same way. $\square$

**Lemma 12.** *For any* $u \in \mathcal{S}^{d-1}$, $i \in [n]$, $Q \in \mathcal{O}(d)$, *we have* $y_i \in \mathcal{C}_{\mathcal{Y}}(u,\delta) \implies x_{\pi^\star(i)} \in \mathcal{C}_{\mathcal{X}}(Q^\top u, \delta + \delta_i(Q,u))$, *where* $\delta_i(Q,u) = 2\sigma \frac{|\langle z_i, (Q^\star)^\top u \rangle|}{\|x_{\pi^\star(i)}\|} + \rho(Q^\star - Q)$.

*Proof.* Assume that $y_i \in \mathcal{C}_{\mathcal{Y}}(u,\delta)$. As in the proof of the previous proposition, this reads as:

$$\langle x_{\pi^\star(i)}, (Q^\star)^\top u \rangle \geqslant (1-\delta)\|Q^\star x_{\pi^\star(i)} + \sigma z_i\| - \sigma\langle z_i, (Q^\star)^\top u \rangle,$$

and thus,

$$
\begin{aligned}
\langle x_{\pi^\star(i)}, Q^\top u \rangle &\geqslant \langle x_{\pi^\star(i)}, (Q^\top - (Q^\star)^\top)u \rangle + (1-\delta)\|Q^\star x_{\pi^\star(i)} + \sigma z_i\| - \sigma\langle z_i, (Q^\star)^\top u \rangle \\
&\geqslant -\|x_{\pi^\star(i)}\|\rho(Q - Q^\star) + (1-\delta)\|Q^\star x_{\pi^\star(i)} + \sigma z_i\| - \sigma\langle z_i, (Q^\star)^\top u \rangle \\
&\geqslant -\|x_{\pi^\star(i)}\|\rho(Q - Q^\star) + (1-\delta - \frac{|\langle z_i, (Q^\star)^\top u \rangle|}{\|x_{\pi^\star(i)}\|})\|x_{\pi^\star(i)}\| \\
&= (1-\delta-\delta_i(Q,u))\|x_{\pi^\star(i)}\|.
\end{aligned}
$$

This concludes the proof. $\square$

# E    Proof of Proposition 1

## E.1    Very-fast sorting-based estimator and equivalence with one step of Frank-Wolfe

We have:
$$\nabla f(D) = 2\left(DX^\top XX^\top X - 2Y^\top YDX^\top X + Y^\top YY^\top YD\right),$$

for any bistochastic matrix $D$, leading to, for $J = \frac{\mathbf{1}\mathbf{1}^\top}{n}$:

$$\nabla f(J) = 2\left(JX^\top XX^\top X - 2Y^\top YJX^\top X + Y^\top YY^\top YJ\right).$$

For any permutation matrix $P$, we have since $J^\top = J$ and $JP = J$:

$$
\begin{aligned}
\langle JX^\top XX^\top X, P \rangle &= \langle X^\top XX^\top X, JP \rangle \\
&= \langle X^\top XX^\top X, J \rangle,
\end{aligned}
$$

and similalry:

$$\langle Y^\top Y Y^\top Y J, P \rangle = \langle J Y^\top Y Y^\top Y, P^\top \rangle$$
$$= \langle Y^\top Y Y^\top Y, J P^\top \rangle$$
$$= \langle Y^\top Y Y^\top Y, J \rangle .$$

Therefore,

$$\arg\min_{P \in \mathcal{S}_n} \langle f(J), P \rangle = \arg\max_{P \in \mathcal{S}_n} \langle Y^\top Y J X^\top X, P \rangle .$$

We have $(X^\top X)_{ij} = \langle x_i, x_j \rangle$ and $(J X^\top X)_{ij} = n \langle \bar{x}, x_j \rangle$. Similarly, $(Y^\top Y J)_{ij} = n \langle \bar{y}, y_i \rangle$, and we have $J^2 = J$. Thus,

$$(Y^\top Y J X^\top X)_{ij} = n^2 \sum_{k=1}^n \langle \bar{y}, y_i \rangle \langle \bar{x}, x_j \rangle ,$$

leading to:

$$\arg\min_{P \in \mathcal{S}_n} \langle f(J), P \rangle = \arg\max_{\pi \in \mathcal{S}_n} \sum_{i \in [n]} \sum_{k=1}^n \langle \bar{y}, y_i \rangle \langle \bar{x}, x_{\pi^\star(i)} \rangle ,$$

and to the following sorting-based estimator, that can be computed very easily in $O(nd \log(n))$ computes.

$$\hat{\pi} \in \arg\max_{\pi \in \mathcal{S}_n} \frac{1}{n} \sum_{i=1}^n \langle x_{\pi(i)}, \bar{x} \rangle \langle y_i, \bar{y} \rangle , \tag{28}$$

where $\bar{x} = \frac{1}{n} \sum_i x_i$ and $\bar{y} = \frac{1}{n} \sum_i y_i$ are the mean vectors of each point cloud. The idea is that thanks to the scalar product, this estimator gets rid of the orthogonal trasformation. Its strength is that it can be computed in $\mathcal{O}(n \log(n))$ iterations, since it consists in sorting two vectors. We have the following result for this estimator.

**Proposition 2.** *Let $\delta \in (0,1)$ and $\varepsilon > 0$. If $\sigma \ll n^{\frac{12(1+2\varepsilon)}{\delta}}$, the estimator $\hat{\pi}$ as defined in Equation (28) satisfies with high probability:*
$$\mathrm{ov}(\hat{\pi}, \pi^\star) \geqslant 1 - \delta .$$

*Proof.* Without loss of generality, we can assume that $\pi^\star = \mathrm{Id}$. Then, for all $i$,

$$\langle y_i, \bar{y} \rangle = \langle Q^\star x_i + \sigma z_i, Q^\star \bar{x} + \sigma \bar{z} \rangle$$
$$= \langle x_i, \bar{x} \rangle + \sigma^2 \langle z_i, \bar{z} \rangle + \sigma \langle z_i, Q^\star \bar{x} \rangle + \sigma \langle Q^\star x_i, \bar{z} \rangle ,$$

so that:

$$\frac{1}{n} \sum_{i=1}^n \langle x_{\pi(i)}, \bar{x} \rangle \langle y_i, \bar{y} \rangle = \frac{1}{n} \sum_{i=1}^n \langle x_{\pi(i)}, \bar{x} \rangle \langle x_i, \bar{x} \rangle + \frac{1}{n} \sum_{i=1}^n \langle x_{\pi(i)}, \bar{x} \rangle \left[ \sigma^2 \langle z_i, \bar{z} \rangle + \sigma \langle z_i, Q^\star \bar{x} \rangle + \sigma \langle Q^\star x_i, \bar{z} \rangle \right]$$

$$= -\frac{1}{2n} \sum_{i=1}^n \langle x_{\pi(i)} - x_i, \bar{x} \rangle^2 + \frac{1}{n} \sum_{i=1}^n \langle x_i, \bar{x} \rangle^2 \langle x_i, \bar{x} \rangle$$

$$+ \frac{1}{n} \sum_{i=1}^n \langle x_{\pi(i)}, \bar{x} \rangle \left[ \sigma^2 \langle z_i, \bar{z} \rangle + \sigma \langle z_i, Q^\star \bar{x} \rangle + \sigma \langle Q^\star x_i, \bar{z} \rangle \right] .$$

By definition of $\hat{\pi}$, we have $\frac{1}{n} \sum_{i=1}^n \langle x_{\hat{\pi}(i)}, \bar{x} \rangle \langle y_i, \bar{y} \rangle \geqslant \frac{1}{n} \sum_{i=1}^n \langle x_i, \bar{x} \rangle \langle y_i, \bar{y} \rangle$, that thus writes as:

$$\frac{1}{2n} \sum_{i=1}^n \langle x_{\pi(i)} - x_i, \bar{x} \rangle^2 \leqslant \frac{1}{n} \sum_{i=1}^n \langle x_{\hat{\pi}(i)} - x_i, \bar{x} \rangle \left[ \sigma^2 \langle z_i, \bar{z} \rangle + \sigma \langle z_i, Q^\star \bar{x} \rangle + \sigma \langle Q^\star x_i, \bar{z} \rangle \right]$$

$$\leqslant \sup_{i \in [n]} |\langle x_{\hat{\pi}(i)} - x_i, \bar{x} \rangle| \times \frac{1}{n} \sum_{i=1}^n \left| \sigma^2 \langle z_i, \bar{z} \rangle + \sigma \langle z_i, Q^\star \bar{x} \rangle + \sigma \langle Q^\star x_i, \bar{z} \rangle \right| .$$

We will first bound this right hand side. First, for all $i, j \in [n]$, $x_i - x_j$ is independent from $\bar{x}$, so that conditionally on $\bar{x}$ we have $\langle x_i - x_j, \bar{x} \rangle \sim \mathcal{N}(0, 2\|\bar{x}\|^2)$, leading to:

$$\mathbb{P}\left( |\langle x_i - x_j, \bar{x} \rangle| > t \|\bar{x}\| \right) \leqslant 2 \exp(-t^2/2) ,$$

and
$$\mathbb{P}\left(\forall i,j\in[n],|\langle x_i-x_j,\bar{x}\rangle|>t\|\bar{x}\|\right)\leqslant 2\exp(-t^2/2+2\log(n))\,,$$
so that with probability $1-2/n^2$, $\sup_{i,j}|\langle x_i-x_j,\bar{x}\rangle|\leqslant 2\sqrt{2\log(n)}\|\bar{x}\|$. Similarly, with probability $1-4/n^2$, $\sup_i|\langle z_i,Q^\star\bar{x}\rangle|\leqslant 2\sqrt{\log(n)}\|\bar{x}\|$ and $\sup_i|\langle Q^\star x_i,\bar{z}\rangle|\leqslant 2\sqrt{\log(n)}\|\bar{z}\|$.

Then, we can write $z_i=z_i'+\bar{z}$ where $z_i'$ is Gaussian (its covariance matrix is the projection on the orthogonal of $\bar{z}$) and independent from $\bar{z}$. Thus, $\sup_i|\langle z_i,\bar{z}\rangle|\leqslant\|\bar{z}\|^2+\sup_i|\langle z_i',\bar{z}\rangle|\leqslant\|\bar{z}\|^2+2\sqrt{\log(n)}\|\bar{z}\|$ with probability $1-2/n^2$.

Thus, with probability $1-8/n^2$ and for $\sigma\leqslant 1$,
$$\sup_{i\in[n]}|\langle x_{\hat{\pi}(i)}-x_i,\bar{x}\rangle|\times\frac{1}{n}\sum_{i=1}^n\left|\sigma^2\langle z_i,\bar{z}\rangle+\sigma\langle z_i,Q^\star\bar{x}\rangle+\sigma\langle Q^\star x_i,\bar{z}\rangle\right|$$
$$\leqslant 2\sqrt{2\log(n)}\sigma\|\bar{x}\|\left[2\sqrt{\log(n)}\|\bar{x}\|+\|\bar{z}\|^2+4\sqrt{\log(n)}\|\bar{z}\|\right]$$

Now, $n\|\bar{x}\|^2$ and $n\|\bar{z}\|^2$ are both $\chi_d^2$ random variables, so that
$$\mathbb{P}\left(\max(|n\|\bar{x}\|^2-d|,|n\|\bar{z}\|^2-d|)>2t+2\sqrt{dt}\right)\leqslant 4e^{-t}\,.$$

For $t=2(\sqrt{2}-1)d$, this leads to, with probability $4e^{-2(\sqrt{2}-1)d}$:
$$\|\bar{x}\|^2,\|\bar{z}\|^2\in[1/2,3/2]\frac{d}{n}\,.$$

Thus, with probability $1-8/n^2-4e^{-2(\sqrt{2}-1)d}$,
$$\sup_{i\in[n]}|\langle x_{\hat{\pi}(i)}-x_i,\bar{x}\rangle|\times\frac{1}{n}\sum_{i=1}^n\left|\sigma^2\langle z_i,\bar{z}\rangle+\sigma\langle z_i,Q^\star\bar{x}\rangle+\sigma\langle Q^\star x_i,\bar{z}\rangle\right|$$
$$\leqslant 2\sigma\sqrt{2\log(n)}\|\bar{x}\|^2\left[2\sqrt{\log(n)}+3\sqrt{d/n}+4\sqrt{3\log(n)}\right]$$
$$=C\log(n)\sigma\|\bar{x}\|^2\,,$$

for some numerical constant $C$, if $n\geqslant d$, leading to
$$\frac{1}{2n}\sum_{i=1}^n\langle x_{\hat{\pi}(i)}-x_i,\bar{x}\rangle^2\leqslant C\log(n)\sigma\|\bar{x}\|^2\,.$$

Now, if $i\neq j$ are fixed, for $t\leqslant 1$, $\mathbb{P}\left(\frac{1}{2}\langle x_i-x_j,\bar{x}\rangle^2<t\|\bar{x}\|^2\right)=\mathbb{P}\left(\mathcal{N}(0,1)^2<t\right)\leqslant c\sqrt{t}$, for some constant $c>0$.

We are now going to upper bound the probability of the event $\mathcal{A}=$"there exists $\mathcal{I}\subset[n]$ with $|\mathcal{I}|\geqslant\alpha n$ and $\pi$ a permutation such that *(i)* for all $i\in\mathcal{I}$, $\pi(i)\neq i$, *(ii)* $\{i,\pi(i)\}_{i\in\mathcal{I}}$ form disjoint pairs and *(iii)* for all $i\in\mathcal{I}$, $\frac{1}{2}\langle x_{\hat{\pi}(i)}-x_i,\bar{x}\rangle^2\leqslant\beta\|\bar{x}\|^2$", for some constants $\alpha,\beta\in(0,1)$ to be fixed later. Let $\mathcal{I}$ and $\pi$ be fixed. Since $\pi(i)\neq i$, we have $\mathbb{P}\left(\frac{1}{2}\langle x_i-x_{\pi(i)},\bar{x}\rangle^2>\beta\|\bar{x}\|^2\right)\leqslant 2e^{-\beta/2}$, and using *(ii)* all pairs are independent, leading to:
$$\mathbb{P}\left(\pi,\mathcal{I}\text{ satisfies }\textit{(i)-(ii)-(iii)}\right)\leqslant\mathbb{P}\left(\forall i\in\mathcal{I},\quad\frac{1}{2}\langle x_i-x_{\pi(i)},\bar{x}\rangle^2>\beta\|\bar{x}\|^2\right)$$
$$\leqslant c\sqrt{\beta}\,.$$

Thus, using a union bound over all possible $\mathcal{I}$ and $\pi$, we have that:
$$\mathbb{P}\left(\mathcal{A}\right)\leqslant 2^n n^n e^{-\alpha\beta n/2+\alpha n\log(2)}$$
$$=e^{\log(c\sqrt{\beta})\alpha n+n\log(n)+(1+\alpha)n\log(2)}\,.$$

Now, using what we have proved above, denoting $\mathcal{B}$ the event $\frac{1}{2n}\sum_{i=1}^n\langle x_{\hat{\pi}(i)}-x_i,\bar{x}\rangle^2\leqslant C\log(n)\sigma\|\bar{x}\|^2$, we have $\mathbb{P}(\mathcal{B})\geqslant 1-8/n^2-4e^{-2(\sqrt{2}-1)d}$. Let $\mathcal{C}$ be the event $\{\text{ov}(\hat{\pi},\pi^\star)\leqslant 1-\delta\}$.

Under $\mathcal{C} \cap \mathcal{B}$, we have the existence of $\mathcal{I}' \subset [n]$ such that for all indices $i \in \mathcal{I}'$, $\hat{\pi} \neq i$ and $|\mathcal{I}'| \geqslant \delta n/6$. Now, since then $\frac{1}{2|\mathcal{I}'|} \sum_{i \in \mathcal{I}'} \langle x_{\hat{\pi}(i)} - x_i, \bar{x} \rangle^2 \leqslant \frac{6}{\delta} \times C \log(n)\sigma \|\bar{x}\|^2$, we have that at least half of these indices satisfy $\frac{1}{2}\langle x_{\hat{\pi}(i)} - x_i, \bar{x}\rangle^2 \leqslant \frac{12}{\delta} \times C \log(n)\sigma\|\bar{x}\|^2$: we denote by $\hat{\mathcal{I}}$ the set of these indices. Hence, $\hat{\pi}, \hat{\mathcal{I}}$ satisfy properties *(i)-(ii)-(iii)* with $\alpha = \frac{\delta}{12}$ and $\beta = \frac{12}{\delta} \times C \log(n)\sigma$, leading to (taking these constants for $\mathcal{A}$):

$$\mathbb{P}(\mathcal{B} \cap \mathcal{C}) \leqslant \mathbb{P}(\mathcal{A})$$
$$\leqslant e^{\log(c\sqrt{\beta})\alpha n + n \log(n) + (1+\alpha)n \log(2)}$$
$$= \exp\left(\frac{\delta n \log\left(12C\delta^{-1}\log(n)\sigma\right)}{12} + n \log(n) + 2n \log(2)\right).$$

For this probability to be close to zero, we thus need that $-\frac{\delta n \log\left(12C\delta^{-1}\log(n)\sigma\right)}{12} \geqslant (1+\varepsilon)n \log(n)$, which can be written as:

$$-\log\left(12C\delta^{-1}\log(n)\sigma\right) \geqslant \frac{12(1+\varepsilon)\log(n)}{\delta} = \log\left(n^{\frac{12(1+\varepsilon)}{\delta}}\right),$$

which is satisfied for $\sigma \ll n^{\frac{12(1+2\varepsilon)}{\delta}}$. $\qquad\square$

## E.2 The "Ace" estimator

**Proposition 3** (Ace). *Let $\delta_0 > 0$. Assume that $\left\|\hat{Q} - Q^\star\right\|_F^2 \leqslant 2(1-\delta_0)d$ and $\log n \ll d \ll n$. Then, there exists a constant $C > 0$ such that the estimator $\hat{\pi}$ defined in Equation* (4) *satisfies with probability $1 - 2e^{-d/16} - 2n^{-n}$:*

$$\mathrm{ov}(\hat{\pi}, \pi^\star) = 1 - \frac{C}{\delta_0} \max\left(\sqrt{\frac{d\log(d/\delta_0)}{n} + \frac{\log(n)}{d}}, \frac{d\log(d/\delta_0)}{n} + \frac{\log(n)}{d}\right).$$

In the $n \gg d \gg \log(n)$ regime: as long as we have non negligible error $\left\|\hat{Q} - Q^\star\right\|_F^2 \leqslant 2(1-\varepsilon)d$ (notice that for uniformly random $Q$, we have $\left\|\hat{Q} - Q^\star\right\|_F^2 = 2d$), we recover $\pi^\star$ with $1 - o(1)$ overlap: doing just a tiny bit better than random for $\hat{Q}$ is enough to recover $\pi^\star$.

*Proof of Proposition 3.* In this proof we denote $g(\pi) := \frac{1}{n}\sum_{i=1}^n \langle x_{\pi(i)}, \hat{Q}^\top y_i \rangle$. By definition, $\hat{\pi} \in \arg\max_{\pi \in \mathcal{S}_n} g(\pi)$. Writing $g(\hat{\pi}) \geqslant g(\pi^\star)$ gives

$$\frac{1}{n}\sum_{i=1}^n \langle x_{\hat{\pi}(i)}, \hat{Q}^\top Q^\star x_{\pi^\star(i)} \rangle \geqslant \frac{1}{n}\sum_{i=1}^n \langle x_{\pi^\star(i)}, \hat{Q}^\top Q^\star x_{\pi^\star(i)} \rangle + \frac{\sigma}{n}\sum_{i=1}^n \langle x_{\pi^\star(i)} - x_{\hat{\pi}(i)}, \hat{Q}^\top z_i \rangle. \quad (29)$$

Without loss of generality, we assume $\pi^\star = Id$. The term in the LHS hereabove, for fixed $\hat{\pi}, \hat{Q}$, has expectation $\mathrm{ov}(\hat{\pi}, \pi^\star) \mathrm{Tr}(\hat{Q}^\top Q^\star)$. We are going to compute uniform fluctuations. For some fixed $Q \in \mathcal{O}(d), P \in \mathcal{S}_n$,

$$\sum_{i=1}^n \langle x_{\pi(i)}, \hat{Q}^\top Q^\star x_i \rangle = \tilde{X}^\top M \tilde{X},$$

for $\tilde{X} = (x_1^\top, \ldots, x_n^\top)^\top \in \mathbb{R}^{nd}$ and $M \in \mathbb{R}^{nd \times nd}$ that writes as $M = \tilde{P}^\top \tilde{Q}$, where $\tilde{Q} \in \mathbb{R}^{nd \times nd}$ is block diagonal with blocks equal to $Q$, and $\tilde{P} \in \mathbb{R}^{nd \times nd}$ is a block matrix, with blocks of size $n \times n$ that verify $\tilde{P}_{[ij]} = P_{ij}I_n$. Thus, $\|M\|_{\mathrm{op}} = 1$ and $\|M\|_F^2 = nd$. Using Hanson-Wright inequality,

$$\mathbb{P}\left(\left|\sum_{i=1}^n \langle x_{\pi(i)}, Q x_i \rangle - n\mathrm{ov}(\pi, Id)\mathrm{Tr}(Q)\right| > C(t + \sqrt{nd}t)\right) \leqslant 2e^{-t}.$$

Since $d \geqslant \log(n)$, with probability $1 - e^{-d/16}$ we have $\sup_{i \in [n]} \|x_i\| \leqslant 2\sqrt{d}$ using Chi concentration. We now work conditionally on this event.

Letting $\mathcal{N}_\delta$ be a $\delta-$net of $\mathcal{O}(d)$,

$$\mathbb{P}\left(\forall \pi \in \mathcal{S}_n\,,\,\forall Q \in \mathcal{N}_\delta\,,\,\left|\sum_{i=1}^n \langle x_{\pi(i)}, Q x_i\rangle - \mathrm{nov}(\pi, Id)\,\mathrm{Tr}(Q)\right| > C(t + \sqrt{ndt})\right) \leqslant 2e^{-t + n\log(n) + cd^2\log(1/\delta)}\,.$$

Using the fact that $\sum_{i=1}^n \langle x_{\pi(i)}, Q x_i\rangle$ is $n\sup_{i\in[n]} \|x_i\|^2 = 4nd-$Lipschitz in $Q$, we thus have:

$$\mathbb{P}\left(\forall \pi \in \mathcal{S}_n\,,\,\forall Q \in \mathcal{O}(d)\,,\,\left|\sum_{i=1}^n \langle x_{\pi(i)}, Q x_i\rangle - \mathrm{nov}(\pi, Id)\,\mathrm{Tr}(Q)\right| > 4nd\delta + C(t + \sqrt{ndt})\right) \leqslant 2e^{-t + n\log(n) + cd^2\log(1/\delta)}\,.$$

Setting $\delta = \frac{\varepsilon}{16}$ and $t = 2n\log(n) + cd^2\log(8/\varepsilon)$, with probability $1 - 2n^{-n}$, we get that for all $\pi, Q$,

$$\left|\sum_{i=1}^n \langle x_{\pi(i)}, Q x_i\rangle - \mathrm{nov}(\pi, Id)\,\mathrm{Tr}(Q)\right| \leqslant \frac{nd\varepsilon}{4} + C'(n\log(n) + d^2\log(1/\varepsilon) + \sqrt{nd(n\log(n) + d^2\log(1/\varepsilon))})\,.$$

We can thus write, since $\mathrm{Tr}(\hat{Q}^\top Q^\star) \geqslant \varepsilon d$:

$$\frac{1}{n}\sum_{i=1}^n \langle x_{\hat\pi(i)}, \hat{Q}^\top Q^\star x_{\pi^\star(i)}\rangle \leqslant \mathrm{Tr}(\hat{Q}^\top Q^\star)\mathrm{dov}(\pi, Id) + \frac{\varepsilon d}{4} + C'(\log(n) + \frac{d^2\log(1/\varepsilon)}{n} + \sqrt{d(\log(n) + \frac{d^2\log(1/\varepsilon)}{n})})\,.$$

and

$$\frac{1}{n}\sum_{i=1}^n \langle x_{\pi^\star(i)}, \hat{Q}^\top Q^\star x_i\rangle \geqslant \mathrm{Tr}(\hat{Q}^\top Q^\star)d - \frac{\varepsilon d}{4} - C'(\log(n) + \frac{d^2\log(1/\varepsilon)}{n} + \sqrt{d(\log(n) + \frac{d^2\log(1/\varepsilon)}{n})})\,.$$

Similarly than before, we prove that with probability $1 - 2n^{-n}$, we have for all $\pi \in \mathcal{S}_n, Q \in \mathcal{O}(d)$:

$$\left|\sum_{i=1}^n \langle x_{\pi^\star(i)} - x_{\hat\pi(i)}, \hat{Q}^\top z_i\rangle\right| \leqslant \frac{\varepsilon dn}{4} + C'(n\log(n) + d^2\log(1/\varepsilon) + \sqrt{nd(n\log(n) + d^2\log(1/\varepsilon))})\,.$$

Equation (29) thus implies that:

$$\mathrm{Tr}(\hat{Q}^\top Q^\star)\mathrm{dov}(\pi, Id) \geqslant \mathrm{Tr}(\hat{Q}^\top Q^\star)d - \frac{3\varepsilon d}{4} - 3C'(\log(n) + \frac{d^2\log(1/\varepsilon)}{n} + \sqrt{d(\log(n) + \frac{d^2\log(1/\varepsilon)}{n})})\,,$$

leading to:

$$\mathrm{ov}(\pi, Id) \geqslant 1 - \frac{3\varepsilon}{4\,\mathrm{Tr}(\hat{Q}^\top Q^\star)} - \frac{3C'}{\mathrm{Tr}(\hat{Q}^\top Q^\star)}(\frac{\log(n)}{d} + \frac{d\log(1/\varepsilon)}{n} + \sqrt{\frac{\log(n)}{d} + \frac{d\log(1/\varepsilon)}{n}})\,.$$

Setting $\varepsilon = \frac{\delta_0}{d}$ concludes the proof. $\qquad\square$

### E.3   Proof of Proposition 1

*Proof of Proposition 1.* For the first part of Proposition 1, we directly apply Proposition 2 with $\varepsilon = 1/2$ to obtain the result.

For the second part that holds for large dimensions, we apply Proposition 2 for $\varepsilon = 1/2$ and $\delta = 1/8$. Using Lemma 3, the first 'Ping' of **??** 1 leads to $\hat{Q}$ satisfying the assumption of Proposition 3 for some $\delta_0$ bounded away from zero, thus leading to the desired result after the last 'Pong' for $\hat\pi$. $\quad\square$

