# OpenReview forum: "Aligning Embeddings and Geometric Random Graphs: Informational Results and Computational Approaches for the Procrustes-Wasserstein Problem"
_NeurIPS.cc/2024/Conference — NeurIPS 2024 poster_

### Official Review · Reviewer_dVzJ · 2024-07-04

**Soundness:** 2
**Presentation:** 2
**Contribution:** 2
**Rating:** 5
**Confidence:** 3

**Summary:**

The authors consider the alignment problem between a point cloud generated from a Gaussian distribution and its noisy version under orthogonal transformation, formulated as the Procrustes-Wasserstein problem. The authors derive information-theoretic results for both high and low dimensional regime (i.e., bounds on the maximum likelihood estimators to the optimal one). Additionally, the authors also propose an alternating algorithm, namely Ping-Pong algorithm, which is a variant of the algorithm in Grave et al. (2019). Empirically, the authors show advantages of the proposed algorithm over other baselines.

**Strengths:**

+ The authors derive the upper bound (or lower bound) of the maximum likelihood estimator to the optimal one for the Procrustes-Wasserstein (PW) problem in both high and low dimensional regime.
+ The authors propose a variant alternating algorithm, namely Ping-Pong for the PW problem, and illustrate its advantages over other baselines.

**Weaknesses:**

+ The title seems misleading. It is not clear about its relation with random graphs.
+ The motivation of the considered problem seems weak. It is better to elaborate the problem. (e.g., why it is interesting to consider point clouds generated from a Gaussian distribution and its noisy version under orthogonal transformation)
+ It is not clear about the advantage of the proposed Ping-Pong algorithm over the alternating algorithm considered in Grave et al. (2019). It is better to deepen analysis and empirical evidences.

**Questions:**

It is interesting to generalize the problem considered in Kunisky and Niles-Weed (2022) into the considered Procrustes-Wasserstein problem. I have some following concerns:
+ The title seems misleading. Could the authors comment on the relation between the considered problems in line 29-39 with random graphs in the title?
+ Could the authors elaborate why it is interesting to consider Procrustes-Wasserstein problem between a point cloud generated from Gaussian distribution and its noisy version under orthogonal transformation?
+ For the high dimensional regime, why do we need lower-bound on \pi, but upper bound on Q? How’s about the upper-bound on \pi, and lower-bound on Q – do they have some roles in the considered problem setting?
+ For the Algorithm 1, why is it possible to fix T, and K? Does the algorithm converge? Is it possible to have some reasonable stopping conditions?
+ Could the authors clarify why the algorithm 1 is superior to the algorithm in Grave et al. (2019)? It is better to deepen the analysis and illustrate with rigorous empirical evidence? E.g., with large number of data points (e.g., in Figure 1, it shows for n <= 200).
+ In line 104-108, the authors emphasize the difference with other related works. Additionally, as in Figure 1, the authors also consider the case where d=2 and d=5, it is not clear why these approaches are not considered as baselines (as claimed in line 102-103). Could the authors clarify it?
+ When Q*= I, could the authors compare the presented results with those in Kunisky and Niles-Weed (2022)?
+ The Procrustes-Wasserstein problem is a non-convex problem. It is better to consider its initialization problem as well. It is not clear about the importance of the one-step analysis for Ping-pong algorithm in the context of nonconvex optimization? Could the authors clarify it?

---------
The rebuttal has addressed some of my concerns, I increase the score to 5.

**Limitations:**

It seems there is no discussion on the limitation.

---

> ### Author Rebuttal · Authors · 2024-08-05
>
> We thank the reviewer for the time spent reviewing and all the remarks and questions that will help clarify our paper. We make sure to address the concerns raised below.
>
>
> *The title seems misleading. Could the authors comment on the relation between the considered problems in line 29-39 with random graphs in the title?* There is a strong connection between the PW (Procrustes-Wasserstein) problem and GGA (geometric graph alignment), as we argue in our paper.
> In GGA, we observe two complete random graphs and try to recover an underlying node correspondence $P^*$. The randomness in this model comes from edge weights which are induced by a Gaussian model: The weights are scalar products between unobserved gaussian random vectors. Note that applying an orthogonal map $Q^*$ to all the vectors leaves the scalar products invariant.
> In PW on the other hand, we directly observe the vectors and want to recover both $P^*$ and $Q^*$.  As described in section 'Geometric graph alignment' of the introduction, the two problems (GGA and PW) are equivalent. This is formalized in Lemma 1, for which a proof is given in Appendix A.
> To make it clearer, we propose to highlight Lemma 1 more in a revised version since it clarifies the title.
>
>
> *Could the authors elaborate why it is interesting to consider Procrustes-Wasserstein problem between a point cloud generated from Gaussian distribution and its noisy version under orthogonal transformation?*
> This is a fundamental question which can be answered as follows. Many learning problems on real-world data can be phrased as  optimization problems, which are often difficult in their worst case formulation, meaning the case where *any* instance of the problem is considered. A common approach to understanding a problem's hardness and deriving algorithmic guarantees is to consider the *planted version* of these problems, that is when data has more structure. This structure stems from a signal (here: an underlying alignement $P^*$ and an orthogonal transformation $Q^*$), and noise (here: Gaussian noise). The benefits of this approach is that we have a precise understanding of why the problem is easy or difficult based on the signal-to-noise ratio $\sigma$. For a broad introduction on planted problems, we refer to the survey on community detection by C. Moore, https://arxiv.org/pdf/1702.00467, page 5.
>
>
> *For the high dimensional regime, why do we need lower-bound on $\pi$, but upper bound on $Q$? How’s about the upper-bound on $\pi$, and lower-bound on $Q$ – do they have some roles in the considered problem setting?*
> An upper bound on $\ell^2$ (for $Q$) and a lower bound on the overlap of $\pi$  point in the same direction: To show that recovery is possible, a larger overlap is better and a lower $\ell^2$-loss is better as well.
>
>
> *For the Algorithm 1, why is it possible to fix T, and K? Does the algorithm converge? Is it possible to have some reasonable stopping conditions?*
> It is possible to fix $K,T$ arbitrarily, depending on the computation power available.
> Empirically, the algorithm converges after $O(1)$ iterations.
> The smaller the signal-to-noise ratio, the larger the number of required iteration.
> We show in Proposition 1 that if $\sigma$ is small enough, $K,T=1$ is already sufficient.
>
>
> *Could the authors clarify why the algorithm 1 is superior to the algorithm in Grave et al. (2019)? It is better to deepen the analysis and illustrate with rigorous empirical evidence? E.g., with large number of data points (e.g., in Figure 1, it shows for n <= 200)*
> Intuitively, our algorithm is better than that of Grave et al. (2019) since it is more ‘greedy' at each iteration.
> Empirically, Figure 1 shows that this is indeed the case for the whole considered range of parameters. The superiority is even stronger for small signal-to-noise ratios and large dimensions.
> We propose, as suggested by the reviewer, to extend our empirical analysis to larger number of data points ($n$ larger) and larger dimensions.
>
>
> *In line 104-108, the authors emphasize the difference with other related works. Additionally, as in Figure 1, the authors also consider the case where d=2 and d=5, it is not clear why these approaches are not considered as baselines (as claimed in line 102-103). Could the authors clarify it?*
> In figure 1, we consider $d=2$ and $d=10$. These are no baselines; they are respectively meant to capture both the small dimension  ($d << \log(n)$) and large dimension ($d>> \log(n)$) settings, as our paper shows that there is a dfference between these two regimes.
>
>
> *When Qstar = I, could the authors compare the presented results with those in Kunisky and Niles-Weed (2022)?*
> This is the content Table 1: in particular, we show that for small dimensions, since we consider the $c^2$ transport cost, we are able to recover some signal even for $\sigma=\Omega(1)$, whereas Kunisky and Niles-Weed (2022) require much smaller signal-to-noise ratio ($\sigma<<n^{-1/d}$).
> We will make this comparison clearer in the paper.
>
>
> *The Procrustes-Wasserstein problem is a non-convex problem. It is better to consider its initialization problem as well. It is not clear about the importance of the one-step analysis for Ping-pong algorithm in the context of nonconvex optimization? Could the authors clarify it?*
> Due to lack of space, we refer to our detailed answer in the rebuttal to Reviewer ERNU (first part).
>
> *It seems there is no discussion on the limitation.*
> We propose to discuss the limitations of our work more clearly in a new separate paragraph, such as the problems left open, negative lower bounds and sharpness of our results, a deepened analysis of our algorithm and its initialization, the existence of computational-to-statistical gaps, and planted problems as tractable proxies for more complex problems.
>
> We hope that our rebuttal answers the reviewer's main questions. We would greatly appreciate an adjusted review score if the concerns are lifted, and remain available for questions.

---

> > ### Comment · Reviewer_dVzJ · 2024-08-11
> >
> > I thank the authors for your explanation in the rebuttal. It addresses some of my concerns, and I increase the score to 5.
> >
> > I have some quick questions as follows:
> >
> > **(1) For GGA (geometric graph alignment)**
> > - Could you elaborate with further details about applying an orthogonal map $Q^*$ to all the vectors leaves the scalar products invariant? I agree about the role of $P^*$, but it is still unclear about $Q^{*}$ in the GGA?
> > - Do we only try to recover the node correspondence? how's about the role of edge weights in the GGA?
> >
> > **(2) For the proposed Ping-Pong algorithm**
> > - In Grave et al. (2019), they also consider the convex relaxation and leverage the Frank-Wolfe algorithm as in the proposed algorithm. Could you comment about it? It is better in case you could give some analysis/discussion and/or empirical illustration for them?
> >
> > I will further adjust the score accordingly.

---

> > > ### Author Response · Authors · 2024-08-12
> > > **Response and clarifications**
> > >
> > > Thank you for your questions.
> > >
> > > **(1)**
> > > In GGA, whatever the orthogonal transformation applied, the scalar product is unchanged: $\langle Qx_i, Qx_j\rangle = \langle x_i,x_j\rangle$.
> > > We thus only seek to retrieve the permutation, since the GGA problem is invariant by orthogonal transformation.
> > > As for the role of edge weights, it is unclear for us what the question is: our point above is that the edge weights observed are invariant under orthogonal transformations.
> > >
> > > **(2)**
> > > Grave et al indeed use the same initialization.
> > > Our paper distinguishes itself from their work in two ways. The first is in the methodology: the *amplification* step is more greedy and thus more efficient, while having the same computational cost. The second is that we propose a 1-step analysis, paving the way for a more general analysis.

---

> > > > ### Comment · Reviewer_dVzJ · 2024-08-12
> > > >
> > > > Thank you for your answer. I have some follow-up questions
> > > >
> > > > **(1) about Lemma 1**
> > > > - The PW problem is non-convex for both (P, Q). However, in the proof of Lemma 1, line 119, it mention about the (sub)optimal $\pi*$ (which lead to the optimal $Q^*$). It is still unclear how to approximate such $\pi^*$ since the optimization problem involves both $(\pi, Q)$. Could you comment about it?
> > > >
> > > > **(2) about GGA**
> > > > - Is it right that in GGA, we only care about the node corresponding $P^*$. Do we care about $Q^*$ in GGA? I confused about the role of $Q^*$ in GGA, do we really need to use $G$ in GGA?

---

> > > > > ### Author Response · Authors · 2024-08-13
> > > > > **Answers to follow-up**
> > > > >
> > > > > Thank you again for your additional questions which we hope to clarify in the following.
> > > > >
> > > > > Let's start by recalling the two problems (GGA) and (PW). In both cases, we consider two point clouds written into matrices, namely $X = (x_1 | x_2 | .... | x_n)$ and $Y = (y_1 | y_2 | .... | y_n)$ with $x_i, y_i \in \mathbb{R}^d$. In this setting, (GGA) amounts to solving the minimization problem
> > > > >
> > > > > $
> > > > > P^*_{GGA} \in \mathrm{argmin}_{P \in \mathcal{S}_n} \Vert X^\top X P - P Y^\top Y \Vert_F^2
> > > > > $
> > > > >
> > > > > while (PW) amounts to solving
> > > > >
> > > > > $
> > > > > (P^*_{PW}, Q^*_{PW}) \in \mathrm{argmin}_{(P,Q) \in \mathcal{S}_n \times \mathcal{O}_d} \Vert QX - YP \Vert_F^2
> > > > > $
> > > > >
> > > > > Lemma 3 tells us that in (PW), it suffices to retrieve either $P^*_{PW}$ or $Q^*_{PW}$ to retrieve a very good guess of its respective counterpart.
> > > > >
> > > > > If we understand you correctly, the key to both of your questions lies in the following "soft reformulation" of the optimization problem which was performed by (Grave, Joulin, Berthet, 2018) in their subsection **Convex relaxation**. The idea is to consider the following problem which is equivalent to (PW):
> > > > >
> > > > > $
> > > > > \text{max}_{P \in \mathcal{S}_n} \max_Q \text{tr}(Q X P^T Y^T)
> > > > > $
> > > > >
> > > > > One has
> > > > >
> > > > > $
> > > > > \max_Q \text{tr}(Q X P^T Y^T) = \Vert X P^T Y^T \Vert_\star
> > > > > $
> > > > >
> > > > > where
> > > > > $\Vert A \Vert_\star$
> > > > > denotes the trace norm (or nuclear norm) of a matrix
> > > > > $A$.
> > > > > If we put the singular values
> > > > > $\sigma_1, ..., \sigma_m$
> > > > > of
> > > > > $A$
> > > > > into a vector
> > > > > $v := (\sigma_1, ..., \sigma_m)$,
> > > > > then
> > > > > $\Vert A \Vert_\star= \sum_i \sigma_i = \Vert v \Vert_1$.
> > > > > Since
> > > > > $\Vert A \Vert_F = \sqrt{\sum_i \sigma_i^2} = \Vert v \Vert_2$,
> > > > > one can argue, using norm equivalence in
> > > > > $ \mathbb{R}^m$,
> > > > > that the problems
> > > > > $ \max_P \Vert X P^T Y^T \Vert_\star $
> > > > > and
> > > > > $ \max_P \Vert X P^T Y^T \Vert_F^2 $
> > > > > are very similar. [Note: We don't know a result to quantify how similar the solutions are, this is why this whole "soft reformulation" is more about giving intuition than providing rigorous results.]
> > > > >
> > > > > Developing the square in $ \max_P \Vert X P^T Y^T \Vert_F^2 $ and omiting constant terms, this problem is further equivalent to
> > > > >
> > > > > $ \max_P \mathrm{tr}( Y P X^T X P^T Y^T ) = \text{max}_P \mathrm{tr}( X^T X P^T Y^T Y P) $
> > > > >
> > > > > Rewriting this as the minimum of the squared norm of a matrix distance, one obtains exactly (GGA). This whole reformulation process can also be reversed to move back, from (GGA) to (PW).
> > > > >
> > > > >
> > > > > Now, let's move to answering your concrete questions:
> > > > >
> > > > > **(1) about Lemma 1**
> > > > >
> > > > > It is true that in order to infer $Q^*$, one needs a good estimate of $P^*$. To obtain this estimate, we use the above "soft reformulation" and first solve (GGA) which is agnostic to $Q^*$. This is done via convex relaxation of $\mathcal{S}_n$ to the set of doubly stochastic matrices, followed by Frank-Wolve optimisation. This procedure is what Grave et al. call the initialisation to their algorithm; it is also the initialisation to our Ping-Pong steps. The doubly stochastic matrix obtained at convergence of Frank-Wolfe if our initial guess of $P^*$. Our one-step analysis of the algorithm lets us hope that this is indeed a good guess in general.
> > > > >
> > > > > **(2) about GGA**
> > > > >
> > > > > You are right that there is no retrieval of $Q^*$ in GGA. Our hope is that the "soft reformulation" can give a good idea where this $Q^*$ is hidden in the process. Since the geometric graph in GGA only depends on the scalar products between the vectors in $X$ and $Y$, our proof of lemma 1 explains in detail how an orthogonal transformation between the point clouds emerges when doing the problem reformulation (GGA) $\to$ (PW).

---

### Official Review · Reviewer_JfWm · 2024-07-13

**Soundness:** 3
**Presentation:** 3
**Contribution:** 3
**Rating:** 6
**Confidence:** 3

**Summary:**

This paper studies the Procrustes-Wasserstein problem that aims to match two high-dimensional point clouds where one is a noisy version of the other up to an orthogonal transformation. The authors establish information-theoretic results in the high ($d \gg \log n$) and low ($d \ll \log n$) dimensional regimes. Further, the authors also propose a "Ping-Pong algorithm" that alternatively estimates the orthogonal transformation and the matching. Sufficient conditions for the method to recover the planted signal after one step is provided. The theoretical finds are also supported by numerical experiments.

**Strengths:**

1. This paper defines a planted model for the Procrustes-Wasserstein problem that extends the work of Kunisky and Niles-Weed [2022] and Wang et al. [2022].
2. Focusing on the $L_2$ transport cost between the point clouds, in contrast to previous works that mostly consider the overlap, the authors established information-theoretic limits in the low-dimensional regime ($d \ll \log n$), which substantially differ from those of Wang et al. [2022], and in the high-dimensional regime ($d \gg \log n$),
which was not explored before.
3. A "Ping-Pong Algorithm", first initialized by a Franke-Wolfe convex relaxation, then alternatively estimating the orthogonal transformation and the relabeling, is proposed. Statistical guarantees for *one single step* of the algorithm is analyzed.

**Weaknesses:**

1. Due to technical challenges, only guarantees for one step of the proposed algorithm is analyzed.
2. The recovery guarantees are only provided in the overlap metric rather than the $c^2$ loss which is claimed to behave very differently when $d$ is small.
3. The dependency on the noise parameter $\sigma$ in the statistical rates seems to be different from that of the ML estimators. The tightness of the results in terms of $\sigma$ is unknown.

**Questions:**

1. How do the statistical guarantees for the proposed algorithm compare to the information-theoretic results? It would be nice if the authors could remark on this.
2. Are there any negative information-theoretic results, i.e., lower bounds on the costs, for the Procrustes-Wasserstein Problem of the planted model?
3. Can there possibly be computational-statistical gaps for the problem?

**Limitations:**

The authors addressed most limitations of the work. Some additional ones are highlighted in Weaknesses and Questions sections.

---

> ### Author Rebuttal · Authors · 2024-08-05
>
> We appreciate the reviewer's feedback and questions, that will help improve the clarity of the paper. We answer the questions raised below.
>
> *Due to technical challenges, only guarantees for one step of the proposed algorithm is analyzed.*
> This is indeed true, our analysis of the Ping-pong algorithm is indeed limited to a 1-step version. As acknowledged in the paper, we did not manage to analyze the Ping-pong algorithm in a more detailed way for two reasons: *(i)* the initialization is a hard problem to study (as we argue in the beginning of Section 3.2, there are no guarantees for the relaxed QAP) and *(ii)* the alternative minimization steps eluded our analysis so far due to the non-convexity of the minimization steps.
> However, it is to be noted that unlike previous results for relaxed QAP (such as for instance Valdivia and Tyagi, 2023), Proposition 1 offers recovery guarantees for non-null noise (even though the noise is required to be small enough).
> Deepening our understanding of the Ping-pong algorithm is a challenging work in progress.
>
> *The recovery guarantees [of the Ping-pong algorithm] are only provided in the overlap metric rather than the loss which is claimed to behave very differently when $d$ is small.*
> For small dimensions, the overlap metric and the $c^2$ transport cost indeed behave very diffently: a guarantee in terms of $c^2$ cost cannot be translated into a guarantee in terms of overlap.
> However, even for small dimensions, guarantees in terms of overlap metric (as the one of Proposition 1) can be translated in terms of transport cost, since we always have $c^2(\pi,\pi') \leq  \max_{i,j} \Vert x_i-x_j\Vert_2^2 \times (1-\text{overlap}(\pi,\pi'))$.
> For large dimensions, we have $\max_{i,j}\Vert x_i-x_j\Vert_2^2 =O(d)$ whp, while for small dimensions we have $\max_{i,j}\Vert x_i-x_j \Vert_2^2 =O(d\log(n))$ whp.
> We will add such a discussion in a revised version.
>
> *The dependency on the noise parameter in the statistical rates seems to be different from that of the ML estimators.*
> This assertion is true: The ML estimator as in Wang et al., 2022 has guarantees that are expressed in terms of overlap metric, while our guarantees are expressed in terms of transport cost $c^2$, which leads to signal recovery even for noise $\sigma$ that does not tend to 0.
>
> *Are there any negative information-theoretic results, i.e., lower bounds on the costs, for the Procrustes-Wasserstein Problem of the planted model?* and *the tightness of the results in terms of $\sigma$ is unknown.*
> The tightness of our results is still an open question we are working on. Hence, we thank you for your inquiry; this is indeed an interesting point. We think that the IT results in the paper are sharp, at least in small dimensions, and not far from being sharp in high dimensions. We believe that whatever the value of $d$, when $\sigma \not \to 0$, one should be able to show that the optimization problem has many solutions, and that some of them are far from the ground truth, in the $c^2$ as well as the $\ell^2$ sense.
>
> *How do the statistical guarantees for the proposed algorithm compare to the information-theoretic results? It would be nice if the authors could remark on this.*
> The statistical guarantees for the Ping-pong algorithm (Section 3) are weaker than the information-theoretic results of Section 2.
> While in Section 2 we are able to recover some signal (in the $\ell^2$ and $c^2$ sense) as long as $\sigma \not \to 0$, Proposition 1 requires $\sigma\to 0$ at a polynomial rate in $n$ to recover any signal for the Ping-pong algorithm.
> However, experiments suggest that the Ping-pong algorithm still recovers some signal even for $\sigma=\Omega(1)$, suggesting that our analysis is suboptimal, leaving this question open for future works.
>
> *Can there possibly be computational-statistical gaps for the problem?*
> This is a very interesting question, that we did not investigate yet.
> Comparing known computational results (Proposition 1 in our paper; the results from Gong and Li, 2024), there is still a gap between our informational results, and the cited computational results.
> However, we believe that this gap is mostly due to suboptimality of the analyses.
> There might be computational-statistical gaps, but investigating these would require using specific methods, for instance the low-degree methods (Hopkins, 2018, *Statistical Inference and the Sum of Squares Method*) that has proven to be efficient for planted recovery problems like the one we are interested in

---

> > ### Comment · Reviewer_JfWm · 2024-08-12
> > **Reply to rebuttal**
> >
> > I thank the authors for the detailed responses. I maintain my postive rating of the paper.

---

### Official Review · Reviewer_ERNU · 2024-07-13

**Soundness:** 3
**Presentation:** 4
**Contribution:** 3
**Rating:** 8
**Confidence:** 3

**Summary:**

This submission is concerned with the problem of aligning planted graphs. To this end, both a permutation $\pi$ and an orthogonal matrix $Q$ must be estimated from observations of $X$ and $Y$. To evaluate the performance of this approach, it is proposed to measure to measure the error induced by $Q$ in terms of the squared Frobenius norm to the true underlying matrix $Q^{\star}$ (denoted $\ell^2(Q,Q^{\star})$) whereas the error induced by $\pi$ is given by $c^2(\pi,\pi^{\star})=\frac 1 n\sum_{i=1}^n||x_{\pi(i)}-x_{\pi^{\star}(i)}||^2$.

The first contribution of this work is to provide information-theoretic results for this problem. Namely, the simple maximum likelihood estimators for $\hat \pi$ (which can be efficiently computed) and $\hat Q$ (which has a closed form solution) recover $\pi^{\star}$ and $Q^{\star}$ almost exactly in the limit of vanishing noise in the high-dimensional regime ($d\geq 2\log n$). In the low-dimensional regime $d\ll \log n$, it is shown that if the noise is of the order $o(d^{-1/2})$, then estimators $\hat Q,\hat \pi$ can be formulated for which $c^2(
hat \pi,\pi^{\star})=o(d)$ and $\ell^2(\hat Q,Q^{\star})=o(d)$. It is shown, moreover that the good performance of one of the two estimators can be transferred to the estimator in a straightforward (and computationally tractable) manner.

Next, as the PW problem is equivalent to the quadratic assignment problem (QAP) which is known to be NP-hard in general, a convex relaxation of the QAP is considered which consists of maximizing the objective over all bistochastic matrices. The proposed algorithm for approximating the QAP consists of first solving the relaxed problem via the Frank-Wolfe algorithm then using this solution to hot start an alternating minimization procedure for the permutation matrix and the orthogonal matrix. Guarantees for one step of this algorithm are then provided. The paper concludes with some numerical experiments which shows that the proposed Ping-Pong algorithm generally outperforms direct resolution of the relaxed QAP and the method of Grave et al.

**Strengths:**

The paper is well-written and places itself well within the existing literature on this problem.

To my understanding, the high dimensional results are first of their kind whereas the low dimensional results improve on previously known results. The proposed algorithm is also seen to perform best out of the other considered.

**Weaknesses:**

The analysis of the proposed ping-pong algorithm is quite limited. The one-step result provided in Proposition 1 is already an interesting step in the analysis, but  it would be interesting to see a more refined analysis. Given the fact that the QAP problem is NP hard it is clear that we cannot expect particularly strong results in general (e.g. convergence to global minimizer), but it may be possible to discern some properties of the matrices to which the algorithm converges or to provide a bound on the number of iterations required in certain cases.
At the very least it would be useful to recall the per iteration complexity.

**Questions:**

The authors are kindly requested to answer to the above point.

I noted the following typos:
1. line 139: is its probability -> if its probability.
2. line 279: envelop -> envelope.

**Limitations:**

The assumptions are clearly stated in each relevant result.

---

> ### Author Rebuttal · Authors · 2024-08-05
>
> We thank the reviewer for the thorough reading and reviewing, positive assessment, and very positive feedback which will help improve the quality of the paper in its second version.
> We answer the questions raised in the review below.
>
> *The analysis of the proposed ping-pong algorithm is quite limited. The one-step result provided in Proposition 1 is already an interesting step in the analysis, but it would be interesting to see a more refined analysis.*
> Our analysis of the Ping-pong algorithm is indeed limited to a 1-step version. As acknowledged in the paper, we did not manage to analyze the Ping-pong algorithm in a more detailed way for two reasons: *(i)* the initialization is a hard problem to study (as we argue at the beginning of Section 3.2, there are no guarantees for the relaxed QAP) and *(ii)* the alternative minimization steps eluded our analysis so far due to the non-convexity of the minimization steps.
> However, it is to be noted that, unlike previous results for relaxed QAP (such as Valdivia and Tyagi, 2023), Proposition 1 offers recovery guarantees for non-null noise (even though the noise is required to be small enough).
> Deepening our understanding of the Ping-pong algorithm is a challenging work in progress.
>
>
> *Given the fact that the QAP problem is NP-hard it is clear that we cannot expect particularly strong results in general (e.g. convergence to global minimizer), but it may be possible to discern some properties of the matrices to which the algorithm converges or to provide a bound on the number of iterations required in certain cases. At the very least it would be useful to recall the per iteration complexity.*
> We will recall the per iteration complexity of the Ping-pong algorithm in a revised version.
> For the initialization, each of the $T$ steps of Frank-Wolfe algorithm is a LAP that has complexity $O(n^3)$ (which can be reduced to $O(n)$ via entropic regularization at the cost of approximations).
> The SVD in the ‘Ping' step has complexity $O(d^3)$ while the LAP in the ‘Pong' step has complexity $O(n^3)$ (again, entropic regularization can reduce this to $O(n)$).
> The overall complexity of the algorithm is thus $O(Tn^3+ K(n^3+d^3))$ (or $O((K+T)n+ Kd^3)$ if we use entropic regularization).

---

> > ### Comment · Reviewer_ERNU · 2024-08-11
> >
> > I thank the authors for clarifying these points. I concur that a more in-depth study of the ping-pong algorithm is of a great interest, but is likely complicated and would be deserving of a separate paper. The addition of the overall complexity is useful in my opinion.

---

### Official Review · Reviewer_6nbD · 2024-07-23

**Soundness:** 3
**Presentation:** 4
**Contribution:** 3
**Rating:** 7
**Confidence:** 3

**Summary:**

This paper studies the theoretical limits of the Procruste-Wasserstein problem which consists in finding an optimal assigment of cloud of points in Euclidean space up to a global rotation. The paper focuses on the model of a cloud of point perturbed by a global rotation and some gaussian noise. The authors establish theoretical results on the recovery of the optimal rotation and the optimal assigment in two cases of interest: high-dimension/low number of samples and low dimensions/high number of samples. They also propose a Frank-Wolfe type of algorithm with an initial step based on convex relaxation: this algorithm is shown to perform better than other state of the art algorithm.

**Strengths:**

The paper is extremely well-written, it is easy to read. In my opinion, the questions addressed in the paper are of interest to the community interested in geometric alignment. The improvement upon Grave's algorithm seems interesting as well.
The estimator in the low-dimensional case introduced the idea of slicing along one-dimensional subspace and it is remarkable that this idea which was used in other contexts, works well in this low-dimensional problem.

**Weaknesses:**

The theoretical results are interesting for a relatively narrow audience of NeurIPS.

**Questions:**

Why is the conical alignment loss not used in practical experiments? Is it because it gives poor practical performances?

**Limitations:**

not applicable.

---

> ### Author Rebuttal · Authors · 2024-08-05
>
> We appreciate the reviewer's positive feedback and the valuable comments.
> We answer the question raised below.
>
> *Why is the conical alignment loss not used in practical experiments? Is it because it gives poor practical performances?*
> The conical alignment loss is useful for informational results only, and is not used in practical experiments because its computational complexity is very high. The reason is that one has to optimize over an $\varepsilon-$net of asymptotic size $(\sqrt{d}/\varepsilon)^{d^2}$, and each evalutation on this $\varepsilon-$net has complexity $O(pn)$ where $p \geq \mathrm{polylog}(1/\sigma, d)$. This yields a total complexity which is superexponential in $d$.
> In other words, the statistical performances of the conical alignment loss are strong, while its computation efficiency is not. We do however believe that the conical alignment loss could be a path towards more efficient algorithms for this problem, that could seek at approximately minimizing this loss.

---

### Decision · Program_Chairs · 2024-09-25

**Decision:**

Accept (poster)

**Comment:**

This paper studies theoretical results for finding an alignment between two sets of points, where the distribution of the first set of points is the orthogonal transformation of the other set of points, up to some noise. Although there are some concerns about the motivation of the problem to a general audience, all reviewers agree that the theoretical results are novel and meaningful. Reviewer concerns were also addressed over multiple discussions during the rebuttal phase.

I would encourage the authors to address reviewer concerns and make improvements to the overall presentation, to strengthen the potential impact of the work.